# Impact of a Sonochemical Approach to the Structural and Antioxidant Activity of Brown Algae (Fucoidan) Using the Box–Behnken Design Method

Uday Bagale [1,*], Ammar Kadi [1], Artem Malinin [1], Varisha Anjum [1], Irina Potoroko [1] and Shirish H. Sonawane [2]

[1] Department of Food and Biotechnology, South Ural State University, Chelyabinsk 454080, Russia; ammarka89@gmail.com (A.K.); artemmalinin3@gmail.com (A.M.); aniumv@susu.ru (V.A.); irina_potoroko@mail.ru (I.P.)

[2] Department of Chemical Engineering, National Institute of Technology Warangal, Warangal 506004, India; shirish@nitw.ac.in

* Correspondence: bagaleu@susu.ru; Tel.: +7-(351)267-93-80

**Abstract:** A fucoidan discovered in the plant *Fucus vesiculosus*, which lowered the molecular weight of fucoidan, was ideal for its application in the pharmaceutical and food sectors. The aim was to study the impact of ultrasound process parameters on the molecular weight, structure, and antioxidant activity of fucoidan. For optimization of sonochemical process parameters such as temperature, sonication time, and power (intensity), Box–Behnken design (BBD) through the response surface method (RSM) at fixed fucoidan concentrations was compared with a normal process. The outcomes demonstrated that sonochemical treatment significantly decreased molecular weight (Mw) to 318 kDa compared to the control process (815 kDa). Antioxidant activity tests revealed that the sonication treatment significantly increased antioxidant activity (88.9% compared to 65.3% with the control process). Through use of the BBD model, we found that the ideal conditions for degradation of fucoidan were a temperature of 33 °C, sonication time of 40 min, and sonication power of 102.5 W/cm$^2$. Under these conditions, the quadratic model was fitted and the experimental values for Mw and antioxidant activity (318 kDa and 87.4%) were close to the predicated values (316 kDa and 87.9%). According to the findings, sonication treatment is a useful method for lowering fucoidan levels with no observable changes in the monosaccharide units of fucoidan through scanning electron microscope, X-ray diffraction, and Fourier-transform infrared (FTIR) analysis.

**Keywords:** fucoidan; brown algae; BBD; RSM; antioxidant activity; sonochemical process; SEM





## 1. Introduction

Fucoidan is a sulfonated polysaccharide extracted from brown algae that is composed of fucose linked by α-(1→2)-, α-(1→3)-, and/or α-(1→4)-glycosidic bonds. Fucoidan has been reported to have antioxidative, antibacterial, antiviral, antitumor, anticoagulant, and anti-inflammatory properties [1,2]. Fucoidan has higher bioavailability than its parent polysaccharide due to its lower molecular weight, which benefits populations with special nutritional needs such as the elderly, newborns, and athletes. As a result, there is growing interest in using food-derived polysaccharides as a functional food ingredient with potential in both nutraceutical and pharmaceutical applications [3–6]. According to some studies, the structural characteristics of sulfated polysaccharides, specifically their molecular weight, the degree of sulfation, the position of the sulfated groups, glucuronic acid and fucose content, and the position of the sulfated groups, determine their antioxidant properties [4,6–9]. Though the higher molecular weight of fucoidan has restricted its use in the food and pharmaceutical sectors, since lower molecular weight polysaccharides diffuse more easily into biological tissues and bloodstreams than higher molecular weight polysaccharides, they are advantageous to biological activities.

Polysaccharides have been broken down using a variety of techniques, including chemical breakdown, enzymatic hydrolysis, and physical depolymerization [10–13]. High temperatures, hydrogen peroxide, or acids are required for chemical degradations such as acid hydrolysis and free radical degradation [11,12]. These procedures not only take a long time, but they also severely damage structures and create unavoidable chemical waste. Moreover, additional purification is required because of the side products that are produced and the chemicals utilized to start reactions. Due to their moderate reaction conditions and controlled product distribution, enzymatic procedures are generally preferred over chemical processes. Different enzymes can be expensive, which has restricted their industrial use. In addition, there are commercial enzymes that can break down sulfated fucan, though the reason for limited industrial use is the high price [12].

The sonication technique is an efficient, energy-saving, and ecologically sustainable method of preparing and processing polymer particles, particularly when aimed at breaking up masses and lowering the particle size and Mw of polysaccharides. The transient and monitored destruction of organic polysaccharides such as starch, pectin, cellulose, and other natural polysaccharides has been widely explored using sonochemistry. Ultrasound is a mechanical wave having a frequency greater than 20 kHz. Acoustic cavitation, caused by high-intensity low-frequency ultrasonication, generates hot spots with short lifetimes, strong local heating of 5000 °C, pressures of 1000 atm, and heating and cooling rates greater than 1010 K/s. Acoustic cavitation may have a mechanical influence due to the rapid deflation of the cavitation bubbles and the formation of free radicals caused by water dissociation [12,14–16]. Both of these processes would reduce the massive polysaccharides to tiny particles. Ultrasonic degradation could not only effectively improve the physicochemical properties of polysaccharides, but also enhance their bioactivities, including antioxidation, anti-cancer, anti-inflammatory, and hypoglycemic activity. However, mainstream investigation of the ultrasonic degradation of sulfated polysaccharides has concentrated on heterostructural polysaccharides from sea brown algae, which has made it challenging to investigate structural changes during the degradation reaction, such as changes in chain linkage and sulfation pattern. This makes it difficult to determine the sonication degradation's specific mechanism of action [4,5]. Xu et al., 2019 [17] studied the fundamental composition and physicochemical characteristics of natural polysaccharides (GLPs) and degraded polysaccharides (GLPUDs) from Ganoderma lucidum, as well as their antioxidant effects. Particle size distribution and scanning electron microscopy (SEM) results revealed that the ultrasonic approach efficiently reduced the polysaccharides in Ganoderma lucidum. Similar monosaccharide units as those in GLPs were present in GLPUDs, although in different molar ratios. The main structure of polysaccharides had not been altered by ultrasonic degradation, as demonstrated by infrared spectra and NMR. In the meantime, the thermal stability of polysaccharides improved after being treated with ultrasound. Antioxidant activity increases with a lower molecular weight. Li et al., 2022 [18] studied the ultrasound degradation of Genistein Combined Polysaccharide (GCP) and showed that there was a slight structural change alongside a difference in the molar ratio with more antioxidant activity when compared to native polysaccharide. Similarly, observation was made for lentinan ultrasound treatment [19] and Laminaria japonica polysaccharide [20,21], which investigated whether yellow tea polysaccharides of high molecular weight subjected to ultrasound treatment were simpler to break than low molecular weight polysaccharides. However, the basic structures of these polysaccharides remained unchanged by ultrasonication, though breaks occurred upon chain scission to the backbone or side branches.

The optimization of ideal conditions for the ultrasound parameter in the depolymerization of fucoidan was studied through response surface methodology. The response surface method (RSM) is an efficient statistical method for enhancing answers that are influenced by independent variables. It can forecast the consequences of variable interactions on the outcome of a process. It is successfully utilized to improve polyphenol extraction conditions by utilizing a variety of extraction procedures from various materials. A kind of RSM called Box–Behnken design (BBD) minimizes the number of tests needed to identify ideal conditions while illuminating the relationships between process variables [22–25]. In

order to create a quadratic response surface for second-order experiments, BBD is used as an exploratory strategy.

To the best of our knowledge, there have not been many studies on the physical and biological changes that result from ultrasonically degrading polysaccharides, and there are even fewer papers on the sonication degradation of marine sulfated polysaccharides. A previously described fucoidan from brown algae is a linear sulfated polysaccharide with well-repeating tetrasaccharide units. In the current study, we compared a control process through BBD (which was utilized to create combinations of the designed arrays of the independent variable) to castoff sonication to explore the mechanism of degradation in this fucoidan and to further examine the structures, antioxidant capabilities, and degradation mechanisms of the degraded products.

## 2. Materials and Methods

Materials: Fucoidan (Fucoid Power-U) was procured from Haewon Biotech Co., Ltd., Seoul, Korea. Distilled water was used for all experiments and all analyses were performed in triplicate.

Ultrasonic treatment: An ultrasonic processor (Volna UZTA 063/22 OM, Biysk, Russia) with a 10 mm horn microtip was used to conduct the ultrasound procedure. The ultrasonic processor ran at a frequency of 22 kHz and had a maximum power of 630 W. A 1 mg/mL concentration of an aqueous solution of fucoidan was prepared with minor modifications [21]. To sustain a constant temperature, the tube was submerged in a water bath (DC-1006, Safe Corp., Ningbo, Zhejiang, China). An ultrasonic probe was inserted into the solution one centimeter below the top surface of the suspension. Investigations were conducted into the impacts of the following sonication (ultrasonic) parameters: duration (0–60 min), temperature (30 °C, 40 °C, and 50 °C), sample concentration (1 mg/mL), and input power level (50, 75, and 100% of the total input power).

Without ultrasound treatment: The process for preparation of the fucoidan sample was the same as the above section with the following parameters: temperature of 30 °C, 40 °C, and 50 °C; sample concentration of 1 mg/mL); rotation speed of 100 rpm, 150 rpm, and 200 rpm; and rotation time of 15–60 min.

### 2.1. Optimization Via the Response Surface Method (RSM)

We decided to use the three-level, three-factor Box–Behnken design (BBD) to improve the ultrasonic degradation conditions. Three levels (–1, 0, and +1) of temperature, ultrasonic intensity, and time were investigated. The entire experimental strategy for each value was expressed in both its actual and coded forms. There were 17 experiments in total (Tables 1 and 2). The following quadratic model was used to fit the response variable: Z is the result response designed by the model, where X and Y denote sonication time (min), intensity (W/cm$^2$), or temperature (°C), respectively; the regression model always has an intercept along linear coefficients (a, b, or c); d and e are squared coefficients; and f is the interaction coefficient. X and Y represent temperature and W/cm$^2$, respectively.

$$Z = a + bX + cY + dX^2 + eY^2 + fX$$

### 2.2. Determination of Average Molecular Weight (Avg Mw)

GPC was used to calculate the average Mw and polydispersity index (PDI) values using the technique labeled by [21], with specific alterations. The average Mw was determined using a Waters 1525 HPLC system (Waters, Milford, MA, USA) and a TSKGEL 4000 column (Tosoh Bioscience, Tokyo, Japan). A sample solution of 20 μL was inserted and eluted with 0.2 M NaCl at 40 °C for 25 min. The GPC software (V 2.3) was used to analyze the results. Each time, the calibration curves were obtained using standard dextran (Sigma) with different molecular weights.

**Table 1.** Experimental run developed for ultrasound using the response surface method (RSM).

| | | Factor 1 | Factor 2 | Factor 3 | Response 1 | Response 2 | Response 3 | Response 4 |
|---|---|---|---|---|---|---|---|---|
| Std. | Run | A: Temperature | B: Ultrasound Power Input | C: Sonication Time | Molecular Weight | PDI | Particle Size | Antioxidant Activity |
| | | Degree C | W/Cm$^2$ | min | kDa | | nm | % |
| 2 | 1 | 50 | 99 | 37.5 | 428 | 0.8466 | 149.7 | 79.5 ± 0.01 |
| 7 | 2 | 30 | 124.5 | 60 | 287 | 0.87 | 235 | 88.3 ± 0.1 |
| 6 | 3 | 50 | 124.5 | 15 | 456 | 0.7 | 432 | 83.1 ± 0.071 |
| 13 | 4 | 40 | 124.5 | 37.5 | 370 | 1.56 | 939 | 83.4 ± 0.037 |
| 16 | 5 | 40 | 124.5 | 37.5 | 372 | 1.56 | 939 | 83.2 ± 0.404 |
| 10 | 6 | 40 | 150 | 15 | 389 | 0.6 | 435 | 68.4 ± 0.124 |
| 5 | 7 | 30 | 124.5 | 15 | 358 | 0.62 | 801 | 82.3 ± 0.31 |
| 9 | 8 | 40 | 99 | 15 | 387 | 0.951 | 846 | 88.9 ± 0.11 |
| 3 | 9 | 30 | 150 | 37.5 | 375 | 1.2 | 373.5 | 78 ± 0.41 |
| 8 | 10 | 50 | 124.5 | 60 | 438 | 0.8 | 265 | 79.8 ± 0.13 |
| 14 | 11 | 40 | 124.5 | 37.5 | 370 | 1.56 | 939 | 83.2 ± 0.01 |
| 17 | 12 | 40 | 124.5 | 37.5 | 370 | 1.56 | 941 | 83.3 ± 0.02 |
| 1 | 13 | 30 | 99 | 37.5 | 421 | 0.976 | 819 | 83.5 ± 0.091 |
| 11 | 14 | 40 | 99 | 60 | 352 | 0.83 | 325 | 83 ± 0.02 |
| 15 | 15 | 40 | 124.5 | 37.5 | 390 | 1.56 | 941 | 83.2 ± 0.0117 |
| 12 | 16 | 40 | 150 | 60 | 440 | 0.7 | 545 | 88.8 ± 0.0234 |
| 4 | 17 | 50 | 150 | 37.5 | 600 | 0.631 | 105 | 74 ± 0.017 |

**Table 2.** Experimental run developed for the control process using the response surface method (RSM).

| | | Factor 1 | Factor 2 | Factor 3 | Response 1 | Response 2 | Response 3 |
|---|---|---|---|---|---|---|---|
| Std. | Run | A: Temperature | B: Speed | C: Rotation Time | Molecular Weight | Particle Size | Antioxidant Activity |
| | | Degree C | RPM | min | kDa | μm | % |
| 4 | 1 | 50 | 200 | 37.5 | 826 | 24.33 | 50.5 ± 0.011 |
| 17 | 2 | 40 | 150 | 37.5 | 810 | 56.7 | 61 ± 0.007 |
| 14 | 3 | 40 | 150 | 37.5 | 810 | 56.25 | 61 ± 0.13 |
| 1 | 4 | 30 | 100 | 37.5 | 813 | 100.8 | 56.7 ± 0.21 |
| 5 | 5 | 30 | 150 | 15 | 815 | 95.2 | 60 ± 0.22 |
| 13 | 6 | 40 | 150 | 37.5 | 810 | 56.7 | 61.1 ± 0.151 |
| 16 | 7 | 40 | 150 | 37.5 | 810 | 56.33 | 61 ± 0.0771 |
| 3 | 8 | 30 | 200 | 37.5 | 799 | 77.25 | 59 ± 0.073 |
| 9 | 9 | 40 | 100 | 15 | 818 | 59.11 | 58 ± 0.13 |
| 12 | 10 | 40 | 200 | 60 | 815 | 55.11 | 59.5 ± 0.111 |
| 11 | 11 | 40 | 100 | 60 | 813 | 59.22 | 59.5 ± 0.214 |
| 7 | 12 | 30 | 150 | 60 | 805 | 89.66 | 65.3 ± 0.014 |
| 6 | 13 | 50 | 150 | 15 | 837 | 30.15 | 54.5 ± 0.05 |
| 8 | 14 | 50 | 150 | 60 | 836 | 27.6 | 52.3 ± 0.0417 |
| 2 | 15 | 50 | 100 | 37.5 | 827 | 33 | 51.7 ± 0.02 |
| 15 | 16 | 40 | 150 | 37.5 | 809 | 56.29 | 61 ± 0.01987 |
| 10 | 17 | 40 | 200 | 15 | 821 | 61.6 | 58.2 ± 0.0743 |

### 2.3. Antioxidant Activity of Samples in Terms of DPPH

A 0.1 mM solution of 2,2,-diphenyl-picrylhydrazine (DPPH) was made in order to examine the samples' ability to inhibit oxidation [26]. Each sample received 20 μL of sample and 280 μL of a DPPH radical solution to prepare it for antioxidant UV absorbance analysis. After 30 min of dark incubation, samples were tested for UV absorbance at 520 nm.

$$\text{DPPH scavenging activity} = \frac{\text{Control Abs} - \text{Sample Abs}}{\text{Control Abs}} \times 100$$

To create a regression line, the absorbance inhibition percentage was plotted against the sample volume of fucoidan. The ratio of the sample and Trolox slopes allowed for the calculation of Trolox equivalent antioxidant capacity (TEAC).

### 2.4. Scanning Electron Microscope (SEM) Analysis of Fucoidan Samples

Before imaging, the samples were coated in gold through ion sputtering for conductiveness and kept on image mounting slots for assessment of the external morphology of samples at 1–10 kV (Jeol JSM-7001F, Moscow, Russia).

### 2.5. Fourier-Transform Infrared (FTIR)

FTIR scans at a wavelength of 400–4000 cm$^{-1}$ were performed for functional characteristic analysis of native and treated fucoidan.

### 2.6. Determination of Monosaccharide Composition

The compositions of monosaccharides were calculated using the PMP-HPLC method, as previously described. In brief, 2 mg fucoidan was hydrolyzed with 2 M trifluoroacetic acid (TFA) for 8 h at 110 °C. The hydrolyzed samples were dissolved in 0.3 M NaOH (450 L) and a 0.5 M methanol solution of PMP (450 L) at 70 °C for 30 min. Before derivatization, lactose was added to each sample as an internal standard. After cooling to room temperature, the mixture was neutralized with 450 L of HCl (0.3 M) and extracted three times with 1 mL of chloroform. HPLC analyses were carried out on a 5 m, 4.6 mm, and 150 mm Agilent ZORBAX Eclipse XDB-C18 column (Agilent, Santa Clara, CA, USA) at 25 °C with UV detection at 250 nm. The mobile phase consisted of aqueous 0.05 M KH2PO4 (pH 6.9) with 15% (solvent A) and 40% (solvent B) acetonitrile. A gradient of B (8–19%) was applied over a 25 min period.

### 2.7. Determination of Sulfate Content

Ion chromatography was used to determine the amount of sulfate in fucoidan. Under nitrogen, 2 M TFA was used to hydrolyze the 2 mg fucoidan fraction at 110 °C for 8 h. Prior to ion chromatography, the hydrolysate was first dissolved in water and then dried under vacuum. The chromatographic apparatus utilized was a Compact IC 761 (Metrohm) with a suppressor system and a separation column (Shodex IC SI-90E, 250 mm 4.6 mm). For ion chromatographic analysis, 1.2 mL of eluent per minute at a flow rate of 1 mM $Na_2CO_3$ + 4 mM $NaHCO_3$ was used with 2.5% acetone. The standard curve was created using the peak area and the known standard sulfate concentration. After injecting 20 μL of sample, the sulfate content of the samples was identified from the conductivity signal.

## 3. Results and Discussion

In order to optimize the process parameters of ultrasound in comparison to the control process and their impact on the molecular properties of fucoidan, studies were conducted in terms of sonication time, sonication temperature, and sonication intensity.

### 3.1. Sonication Time in Comparison to Stirring Time

Figure 1 depicts the duration-dependent effects of ultrasound on changes in the average Mw and PDI of the degraded fucoidan fractions. During the ultrasonic process, no foam was formed.

**Sonication time:** Figure 1 shows that after 15 min, the average Mw dropped sharply from 845 to 395 kDa, and after 220 min, it progressively leant towards a constant value of 358 kDa. During the first 40 min of the treatments, PDI values, which represent the Mw distribution, rose rapidly from 1.121 to 1.561 and then steadily decreased to 1.228 by the conclusion of 60 min. This is because the shear stress generated by the sudden collapse of cavitation bubbles was primarily responsible for the rupture of the polysaccharide particle by ultrasound. Small particles with smaller chains are undisrupted by shear force, whereas polymers with large chains are preferentially disrupted. As a result, the average Mw of fucoidan reduced dramatically in the first 15 min before slowing as the majority of the fucoidan cracked into small units. Due to the presence of both native and fractured fucoidan molecules, the solution changed from being homogeneous to being inhomogeneous over the first 40 min, which led to a rise in PDI values. After repeated ultrasonic treatment,

more native fucoidans were shattered and moved to the lower molecular weight region, resulting in a narrow Mw distribution. A similar result was observed in work related to polysaccharides under higher sonication time [17,19,20].

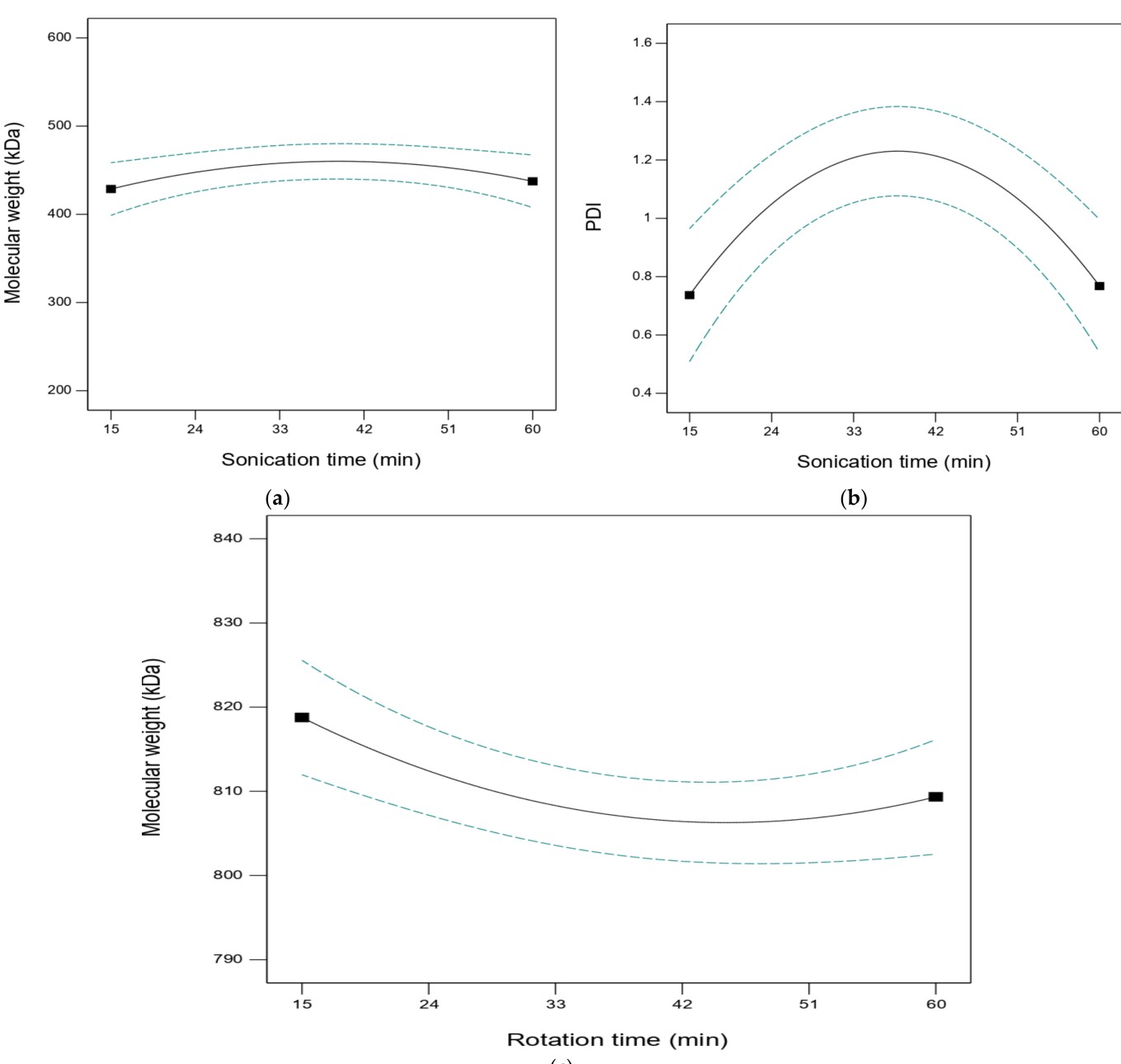

**Figure 1.** (**a**) Molecular weight properties with respect to sonication time, (**b**) polydispersity index with respect to sonication time, and (**c**) control stirring time (ultrasound intensity, 124.5 W/cm$^2$; temperature, 30 °C).

Ultrasound power or intensity. One of the key elements influencing the three separate phases of acoustic cavitation (nucleation, bubble growth, and collapse) is sonication power or intensity. Figure 2a depicts how sonication power affects the average Mw of degraded fucoidan fractions. Under ultrasonic intensities of 99 and 124.5 W/cm$^2$, the average Mw reduced from 845 to 395 and 358 kDa, respectively, in 40 min. The effectiveness of degradation significantly increased as ultrasonic intensity increased, highlighting the significance of sonication power for fucoidan sonolysis ($p < 0.05$). However, higher sonication power may,

within a specific range, intensify the energy of cavitation, reduce the inception of cavitation, and increase the number of cavitation bubbles. This can explain the rapid increase in average Mw from 358 to 421 kDa as the sonication power rose from 124.5 to 150 W/cm$^2$. The PDI values of treated samples also tend to show a reverse trend as they initially increase from 1.1 to 1.27 as sonication intensity increases (99 to 124.5 W/cm$^2$) before later decreasing to 1.17. As higher sonication intensity creates more cavitation bubbles around the acoustic source, this reduces the effectiveness of energy transmission into the reaction medium and slows the ultrasound wave's propagation. As a result, when ultrasonic intensity increased, there was a slight, discernible change in the average Mw and PDI. In comparison to the control process, molecular weight stays similar to its initial value of 825 kDa as there is no attributable change observed, as shown in Figure 2b.

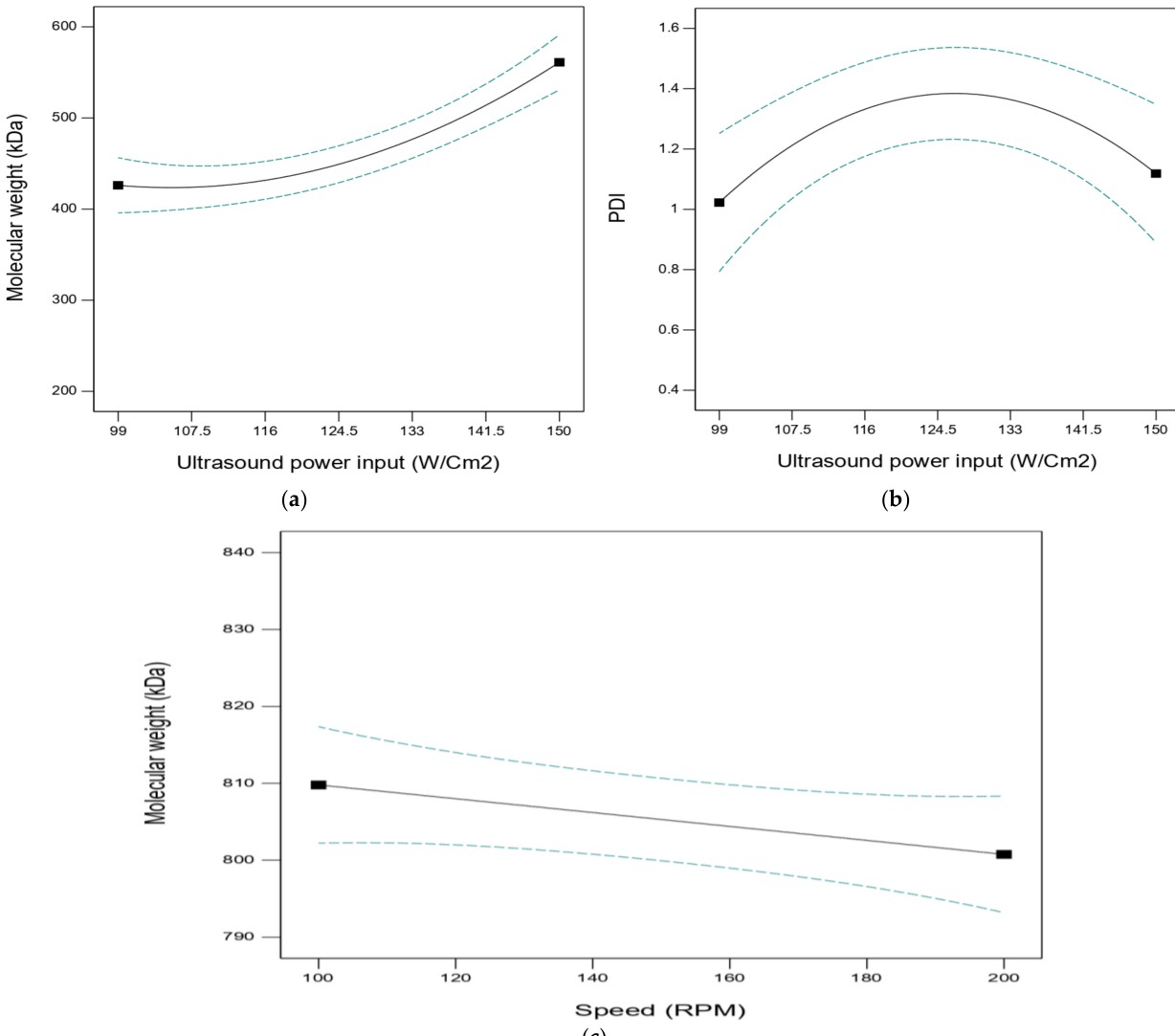

**Figure 2.** (**a**) Molecular weight properties with respect to ultrasound intensity, (**b**) PDI values with respect to ultrasound intensity, and (**c**) control stirring speed (ultrasound time, 40 min; temperature, 30 °C).

Temperature. Figure 3a,c show the effect of temperature on the Mw of fucoidan samples with and without ultrasonic degradation. The findings suggested in Figure 3a suggest that increased temperature would result in less efficient deterioration. After 60 min of treatment, the average Mw of the samples heated to 30 °C, 40 °C, and 50 °C dropped from 850 kDa to 370, 398, and 458 kDa, respectively. This pattern was caused by the growing vapor pressure

associated with the heated liquid, which was a direct result of the reaction temperature rising and made it possible to generate cavitation with a lower sonic intensity. In contrast, as shown in Figure 3c, there was either no or very slight degradation at all temperatures from 30 °C to 50 °C with a molecular weight of 810–845 kDa, as this molecular weight stayed the same as the initial molecular weight and exhibited little difference. The advantage of sonication is that it can intensify the entanglement between polysaccharide chains in solutions even at lower sample concentrations. The molecule's random coiled shape expanded more freely as a result, making it more susceptible to the shear force's attack. Moreover, more diluted fluids have larger velocity gradients near collapsing bubbles, which would aid ultrasonic degradation. Figure 3b shows the PDI of treated samples, showing that increasing temperature gives rise to curvy PDI values (increase–decrease phenomenon in the PDI values).

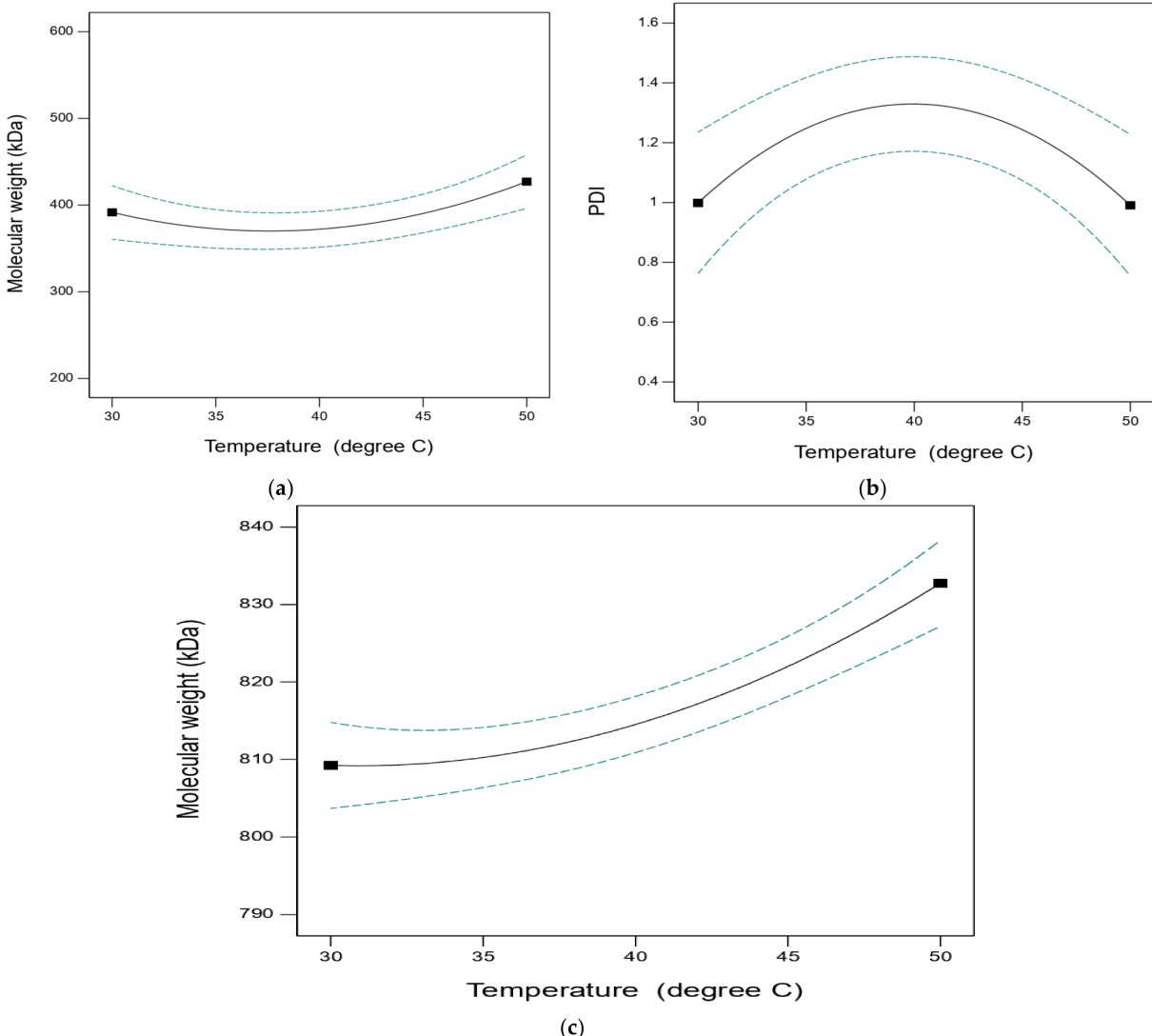

**Figure 3.** (**a**) Molecular weight properties with respect to temperature, (**b**) PDI values with respect to temperature, and (**c**) Mw without sonication (ultrasound time, 40 min; intensity, 124.5 W/cm$^2$).

Impact of Ultrasound or Sonication Parameters on the Particle Size and Antioxidant Activity (AOA) of Treated Samples

Sonication time: Figure 4 shows the effect of sonication time on treated fucoidan in terms of particle size and AOA. The result showed that after 60 min, the average particle size of the samples dropped sharply from 1015 to 257 nm, whereas the antioxidant activity of samples slightly decreased with increased sonication time (from 87% to 85%). This is

due to the fact that the fucoidan sample contained a smaller and larger chain of polymer groups, which led to slight structural changes observed at lower sonication. This represents a drastic change in average particle size that influences the decrease in antioxidant activity to 85%.

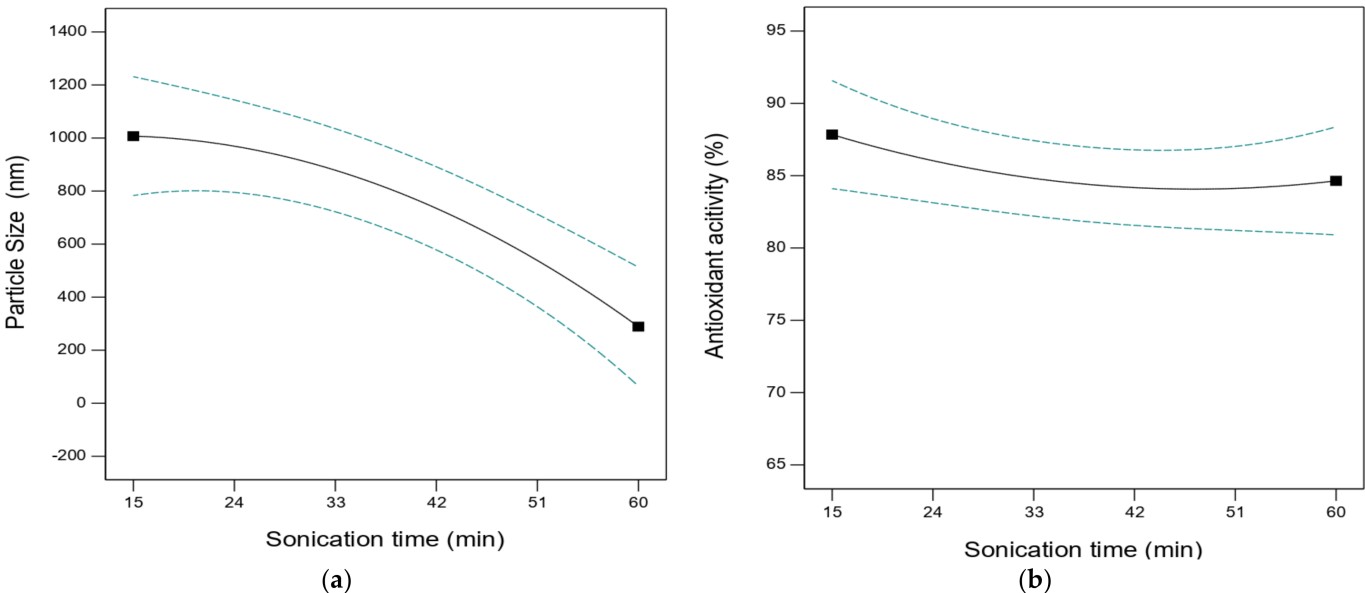

**Figure 4.** (**a**) Particle size and (**b**) AOA of fucoidan in term of sonication time (ultrasound intensity, 124.5 W/cm$^2$; temperature, 30 °C).

Sonication intensity and temperature: Figure 5a–d show the impact of sonication intensity and temperature on particle size and AOA. As shown in Figure 5a,b, as sonication intensity increased from 99 W/cm$^2$ to 124.5 W/cm$^2$, the particle size and AOA increased from 600 to 625 nm and from 83% to 85%, respectively, while further increased sonication intensity decreased particle size and AOA to 395 nm and 81%. The same result can be observed in Figure 5c,d for particle size (600 nm to 197 nm) and AOA (83–79%) with respect to temperature variation (30–50 °C). The reason for the increase in antioxidant activity was that lower Mw polysaccharides would initially have more free hydroxyl and amino groups due to the modest influence of intermolecular hydrogen bonding. Second, at the same mass concentration level, lower Mw polysaccharides contain more reducing sugars. Nevertheless, as temperature and sonication intensity increased, the average Mw could be reduced less quickly, and the polysaccharide structure was more severely destroyed as a result of the increased production of free radicals, which decreased the antioxidant activity. Thus, it is crucial to regulate the duration of ultrasonic treatment in order to regulate molecular size and enhance antioxidant activity [26–29].

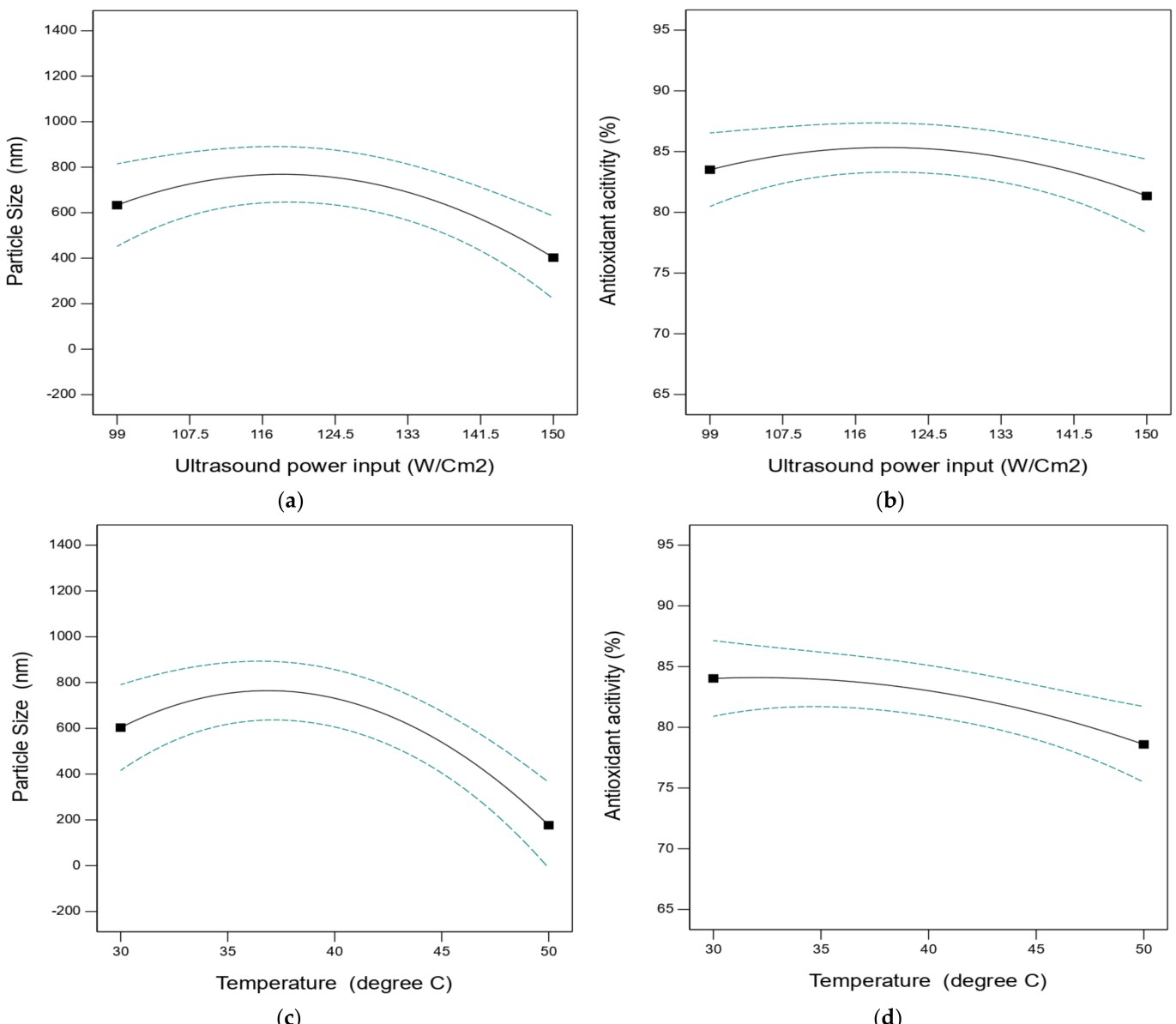

**Figure 5.** (**a**) Particle size and (**b**) AOA in terms of sonication intensity and (**c**) Particle size and (**d**) AOA with respect to temperature.

### 3.2. Optimization by Response Surface Methodology

Based on the single aspect relation amongst the sonication parameters and the properties of the sample, the relationship between independent variables (sonication time, power, and temperature) and response (average Mw, PDI, particle size, and AOA) was further enhanced using RSM on the foundation of three-factor analysis. The results show that molecular weight, PDI values, particle size, and AOA are in the range of 287–600 kDa, 0.6–1.56, 105–941 nm, and 74–88.9% respectively.

Table 3 lists the outcomes of fitting linear, interactive (2FI), quadratic, and cubic models to the experimental data in order to create regression models for samples. The information in Table 4 shows that the quadratic model was best suited to signifying the investigational numbers for fucoidan samples as the values of the quadratic model's determination coefficient ($R^2$), adjusted determination coefficient (adj. $R^2$), and predicated determination coefficient (pred. $R^2$) were the highest and most significant ($p < 0.001$) among all of the models.

**Table 3.** Model summary for all parameters of fucoidan on the basis of BBD.

| Source | Std. Dev. | $R^2$ | Adjusted $R^2$ | Predicted $R^2$ | Lack of Fit $p$-Value | Remarks |
|---|---|---|---|---|---|---|
| **Molecular weight** | | | | | | |
| Linear | 50.63 | 0.5152 | 0.4034 | 0.0362 | 0.0010 | |
| 2FI | 43.46 | 0.7252 | 0.5603 | −0.2002 | 0.0016 | |
| **Quadratic** | **16.40** | **0.9726** | **0.9374** | **0.7661** | **0.0471** | **Suggested** |
| Cubic | 8.76 | 0.9955 | 0.9821 | | | Aliased |
| **PDI** | | | | | | |
| Linear | 0.4136 | 0.0433 | −0.1775 | −0.4218 | 0.0010 | |
| 2FI | 0.4645 | 0.0718 | −0.4852 | −1.3549 | 0.0006 | |
| **Quadratic** | **0.1253** | **0.9527** | **0.8920** | **0.7438** | **0.0223** | **Suggested** |
| Cubic | 0.0000 | 1.0000 | 1.0000 | | | Aliased |
| **Particle size** | | | | | | |
| Linear | 301.00 | 0.2653 | 0.0958 | −0.1698 | <0.0001 | |
| 2FI | 315.96 | 0.3773 | 0.0037 | −0.6695 | <0.0001 | |
| **Quadratic** | **99.27** | **0.9570** | **0.9017** | **0.7116** | **<0.0001** | **Suggested** |
| Cubic | 1.10 | 1.0000 | 1.0000 | | | Aliased |
| **Antioxidant activity** | | | | | | |
| Linear | 4.57 | 0.3562 | 0.2076 | −0.3293 | <0.0001 | |
| 2FI | 2.78 | 0.8171 | 0.7073 | 0.1366 | <0.0001 | |
| **Quadratic** | **1.65** | **0.9546** | **0.8963** | **0.7049** | **<0.0001** | **Suggested** |
| Cubic | 0.0894 | 0.9999 | 0.9997 | | | Aliased |

**Table 4.** ANOVA for results from the quadratic model of molecular weight.

| Source | Sum of Squares | Mean Square | F-Value | $p$-Value | |
|---|---|---|---|---|---|
| **Molecular weight** | | | | | |
| **Model** | 66,857.02 | 7428.56 | 27.61 | 0.0001 | significant |
| A-Temperature | 28,920.13 | 28,920.13 | 107.48 | <0.0001 | |
| B-Ultrasound power input | 5832.00 | 5832.00 | 21.68 | 0.0023 | |
| C-Sonication time | 666.13 | 666.13 | 2.48 | 0.1596 | |
| AB | 11,881.00 | 11,881.00 | 44.16 | 0.0003 | |
| AC | 702.25 | 702.25 | 2.61 | 0.1502 | |
| BC | 1849.00 | 1849.00 | 6.87 | 0.0343 | |
| $A^2$ | 5818.87 | 5818.87 | 21.63 | 0.0023 | |
| $B^2$ | 8309.81 | 8309.81 | 30.88 | 0.0009 | |
| $C^2$ | 3029.81 | 3029.81 | 11.26 | 0.0122 | |
| **Residual** | 1883.45 | 269.06 | | | |
| Lack of fit | 1576.25 | 525.42 | 6.84 | 0.0471 | significant |
| Pure error | 307.20 | 76.80 | | | |
| **Cor total** | 68,740.47 | | | | |
| **Std. dev.** | 16.40 | | | | |

**Table 4.** *Cont.*

| Source | Sum of Squares | Mean Square | F-Value | *p*-Value | |
|---|---|---|---|---|---|
| **Mean** | 400.18 | | | | |
| **C.V. %** | 4.10 | | | | |
| **Adeq precision** | 23.484 | | | | |
| | | | PDI | | |
| **Model** | 2.21 | 0.2461 | 15.68 | 0.0008 | significant |
| A-Temperature | 0.0592 | 0.0592 | 3.77 | 0.0432 | |
| B-Ultrasound power input | 0.0279 | 0.0279 | 1.78 | 0.0241 | |
| C-Sonication time | 0.0135 | 0.0135 | 0.8620 | 0.00841 | |
| AB | 0.0483 | 0.0483 | 3.08 | 0.01228 | |
| AC | 0.0056 | 0.0056 | 0.3584 | 0.1683 | |
| BC | 0.0122 | 0.0122 | 0.7779 | 0.0170 | |
| $A^2$ | 0.4716 | 0.4716 | 30.05 | 0.0009 | |
| $B^2$ | 0.4097 | 0.4097 | 26.10 | 0.0014 | |
| $C^2$ | 0.9613 | 0.9613 | 61.25 | 0.0801 | |
| **Residual** | 0.1099 | 0.0157 | | | |
| Lack of fit | 0.1099 | 0.0366 | | | |
| Pure error | 0.0000 | 0.0000 | | | |
| **Cor total** | 2.32 | | | | |
| **Std. dev.** | 0.1253 | | | | |
| **Mean** | 1.03 | | | | |
| **C.V. %** | 10.15 | | | | |
| **Adeq precision** | 19.8368 | | | | |
| | | | Particle size | | |
| **Model** | $1.534 \text{ E}^{+06}$ | $1.705 \times 10^5$ | 17.30 | 0.0005 | significant |
| A-Temperature | $2.038 \times 10^5$ | $2.038 \times 10^5$ | 20.68 | 0.0026 | |
| B-Ultrasound power input | 58,004.18 | 58,004.18 | 5.89 | 0.0457 | |
| C-Sonication time | $1.636 \times 10^5$ | $1.636 \times 10^5$ | 16.60 | 0.0047 | |
| AB | 40,160.16 | 40,160.16 | 4.08 | 0.0833 | |
| AC | 39,800.25 | 39,800.25 | 4.04 | 0.0844 | |
| BC | 99,540.25 | 99,540.25 | 10.10 | 0.0155 | |
| $A^2$ | $4.903 \times 10^5$ | $4.903 \times 10^5$ | 49.76 | 0.0002 | |
| $B^2$ | $2.360 \times 10^5$ | $2.360 \times 10^5$ | 23.95 | 0.0018 | |
| $C^2$ | $1.150 \times 10^5$ | $1.150 \times 10^5$ | 11.68 | 0.0712 | |
| **Residual** | 68,975.56 | 9853.65 | | | |
| Lack of fit | 68,970.76 | 22,990.25 | 19,158.54 | <0.0001 | significant |
| Pure error | 4.80 | 1.20 | | | |
| **Cor total** | $1.603 \times 10^6$ | | | | |
| **Std. dev.** | 99.27 | | | | |
| **Mean** | 590.01 | | | | |
| **C.V. %** | 06.82 | | | | |
| **Adeq precision** | 19.8860 | | | | |
| | | | AOA | | |

**Table 4.** *Cont.*

| Source | Sum of Squares | Mean Square | F-Value | *p*-Value | |
|---|---|---|---|---|---|
| **Model** | 402.95 | 44.77 | 16.36 | 0.0007 | significant |
| A-Temperature | 30.81 | 30.81 | 11.26 | 0.0122 | |
| B-Ultrasound power input | 82.56 | 82.56 | 30.16 | 0.0009 | |
| C-Sonication time | 36.98 | 36.98 | 13.51 | 0.0079 | |
| AB | 0.0000 | 0.0000 | 0.0000 | 0.0070 | |
| AC | 21.62 | 21.62 | 7.90 | 0.0761 | |
| BC | 172.92 | 172.92 | 63.18 | <0.0001 | |
| $A^2$ | 12.24 | 12.24 | 4.47 | 0.0723 | |
| $B^2$ | 33.13 | 33.13 | 12.10 | 0.0103 | |
| $C^2$ | 13.95 | 13.95 | 5.10 | 0.0586 | |
| **Residual** | 19.16 | 2.74 | | | |
| Lack of fit | 19.13 | 6.38 | 796.98 | <0.0001 | significant |
| Pure error | 0.0320 | 0.0080 | | | |
| **Cor total** | 422.11 | | | | |
| **Std. dev.** | 1.65 | | | | |
| **Mean** | 81.99 | | | | |
| **C.V. %** | 2.02 | | | | |
| **Adeq precision** | 15.4271 | | | | |

Thus, from the above data, we correlated dependent variables against independent variables in the form of regression equations. The regression calculation in relationships of the coded value of the variables was designed as follows.

For molecular weight, the actual equation is predicted by BBD:

$$Z = 2924.18 - 52.5447\ A - 25.9071\ B - 3.4524\ C + 0.2137\ AB + 0.059\ AC + 0.0374728\ BC + 0.371\ A^2 + 0.06832\ B^2 - 0.0529877\ C^2 \tag{1}$$

For PDI

$$Z = -13.9402 + 0.319042A + 0.130757B + 0.0672944\ C - 0.00043098\ AB - 0.000166667AC + 9.62963 \times 10^{-5}\ BC - 0.00334675A^2 - 0.0004797B^2 - 0.000943852\ C^2 \tag{2}$$

For particle size

$$Z = -5424.9 + 191.494A + 61.2913B - 33.8307C + 0.392941AB + 0.443333AC + 0.274946BC - 3.4125\ A^2 - 0.364091B^2 - 0.326519C^2 \tag{3}$$

AOA

$$Z = 52.1254 + 1.55525A + 0.518399B - 1.18747C + 1.260377AB - 0.0103333AC + 0.0114597\ BC - 0.01705A^2 - 0.00431373B^2 + 0.00359506C^2 \tag{4}$$

Z signifies the average Mw, PDI, particle size, and AOA of the degraded fucoidan, respectively, as a function of temperature (A), sonication power (B), and sonication time (C).

Analysis of variance (ANOVA) was carried out to assess the model's applicability and significance, and the results are shown in Table 4. The result shows that the models are highly significant ($p < 0.05$) for all coefficients of regression for Mw, PDI, AOA, and particle size, excluding AC and $C^2$. The developed models can accurately predict the experimental data because the values of $R^2$ and adj $R^2$ are superior to 0.80 and 0.70, respectively. From Table 4, the model F-values are significant for the samples. The lack of fit F-values of samples for Mw, particle size, and AOA are 6.84, 19,158.5, and 796.98, which implies the lack of fit is significant. That means a significant lack of fit is good for validating model

fitting (the value of the lack of fit F-value is shown in Table 4). The signal-to-noise ratio is determined precisely. A ratio greater than four is optimal. Furthermore, the necessary precision for the replies (Table 4) is greater than 13, indicating that the signals are sufficient. According to Table 4, the model's C.V. is less than 10, indicating that it accurately describes the response and that the experimental data are connected with high precision. This means that the predicted value predicted the validated model for the treatment sample.

### 3.3. Impact on Ultrasound Process on Molecular Weight and PDI of Degraded Fucoidan

To demonstrate the interactive effect of independent variables, 3D plots for the response variables (Figure 6) were created using Equations (1) and (2) (ultrasound time, intensity, and temperature). In Figure 6a–c, increasing ultrasound intensity and temperature decreased the molecular weight, while further increasing ultrasound intensity and temperature increased the molecular weight (as already discussed in Figures 1–3). The linear terms of sonication intensity and temperature were significant ($p < 0.0023$ and $p < 0.0001$, respectively), as shown in Table 4 through their interaction and quadratic terms. Hence, as indicated in Figure 6a,c,e, there is a curvilinear decrease or increase in molecular weight for ultrasound intensity and temperature, respectively, while for sonication time, a curvilinear decrease is used. Figure 6b,d,f illustrate the 3D surface plots of the PDI of the degraded fucoidan model with respect to the ultrasound parameter. The interaction terms of ultrasound intensity, time, and temperature were statistically significant ($p < 0.0001$) (Table 4), which resulted in a curvilinear change in the PDI for all parameters investigated. Figure 6 shows that the model's F-value of 15.68 implies the model is significant. There is only a 0.08% chance that an F-value this large could occur due to noise. $p$-values less than 0.0500 indicate that model terms are significant. In this case, $A^2$, $B^2$, and $C^2$ are significant model terms.

### 3.4. Impact of Ultrasound Processes on the Particle Size and AOA of Degraded Fucoidan

Figure 7a–f show the particle size and AOA response variables as a function of the sonication process, with increasing ultrasound intensity and temperature decreasing the particle size and AOA. In the mean, increasing sonication time decreases the particle size and increases AOA. The linear terms of sonication intensity, time, and temperature were significantly less than ($p < 0.005$), while their interaction and quadratic terms were significant. Hence, as demonstrated in Figure 7a–f, there is a curvilinear decrease–increase trend in particle size and AOA.

The ideal circumstances for obtaining response surface plots illustrating the interactions between process factors (sonication time, power, and temperature) and responses (molecular weight kDa, PDI, particle size, nm, and AOA%) are shown in Figures 6 and 7. The ideal conditions for treated fucoidan were a sonication time of 40 min, intensity of 102.76 W/cm$^2$ and a temperature of 33 °C, which led to lower molecular weight (316 kDa) and higher antioxidant activity (87.8%) with lower particle size (567 nm).

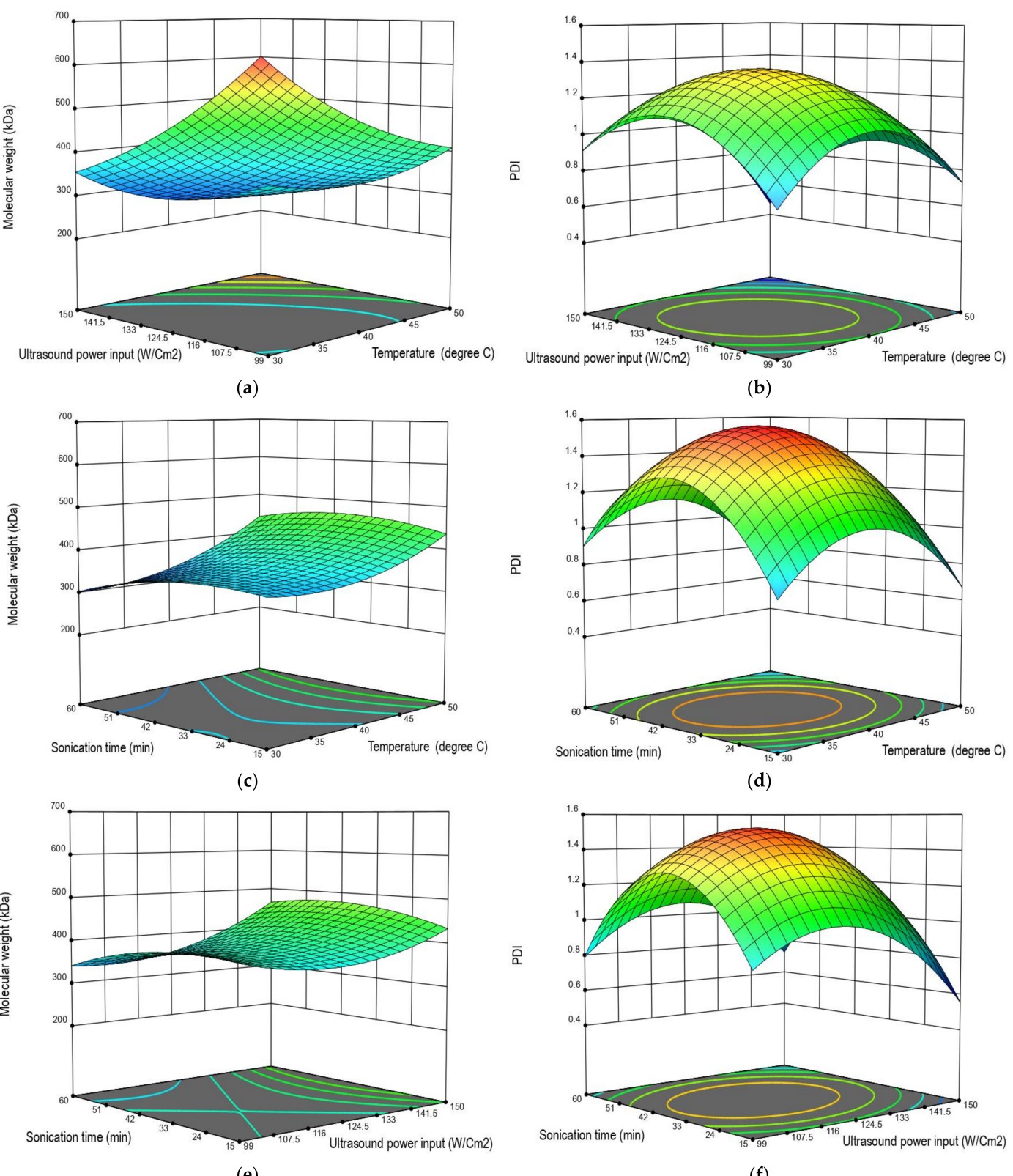

**Figure 6.** Molecular weight (**a**,**c**,**e**) and PDI (**b**,**d**,**f**) as a function of ultrasound parameters.

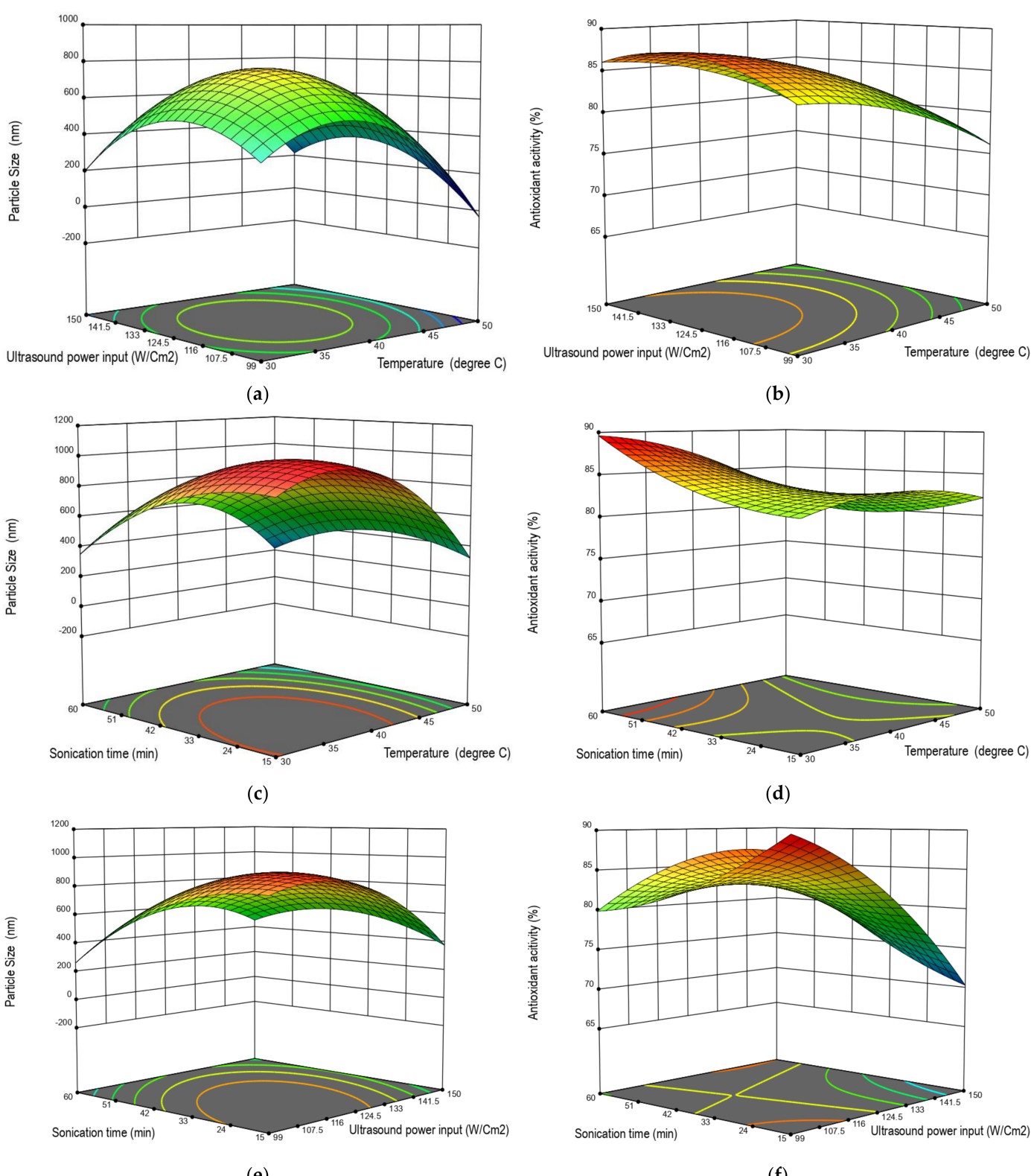

**Figure 7.** Three-dimensional plots for particle size (**a**,**c**,**e**) and AOA (**b**,**d**,**f**) in terms of ultrasound parameters.

*3.5. Validate the Model*

The trials were conducted in the ideal conditions produced by Box–Behnken experiments in order to validate the model. Based on the optimized conditions of the experimental part, we replicated the condition process parameters (sonication time 40 min, power

102.67 W/cm$^2$, and temperature 33 °C) and found them to be near the same response result: molecular weight was 318 kDa, PDI was 1.13, particle size was 568.88, and AOA was 87.44%.

### 3.6. Structure Analysis of Degraded Fucoidan Fractions through FITR and XRD

Fucoidan from the seaweed Kombu contains a straightforward linear backbone and well-repeated units of tetrasaccharides as its structural makeup. These characteristics made it easier for us to investigate the sample's structural changes during ultrasonic treatments and potential deterioration mechanisms. FTIR spectra: Fourier-transform infrared results for ultrasound-treated fucoidan and native fucoidan are shown in Figure 8. The absorption peaks were as follows: the signal at 2925 cm$^{-1}$ was caused by CH stretching of CH$_2$ groups; the absorption at 1650 cm$^{-1}$ arose from stretching vibrations of C=O and C=N; the feature at 1545 cm$^{-1}$ was caused by the bending vibration of -NH; the absorption at 1461 cm$^{-1}$ was caused by the stretching vibration of C=O from the carboxyl group and the bending vibration of OH; the signal at 1228 cm$^{-1}$ was caused by S=O stretching; bands around 1044 cm$^{-1}$ were assigned to C−O stretching vibrations (C-O-H, C-O-C); absorption at 830 cm$^{-1}$ was attributed to C-O-S bending vibrations; and the feature at 560 cm$^{-1}$ was assigned to the asymmetric and symmetric O=S=O deformation of sulfates. From Figure 8, as seen in the clearly visible characteristic peaks at 560 cm$^{-1}$, 1228 cm$^{-1}$, and 830 cm$^{-1}$ caused by S=O and C-O-S, no obvious changes were observed in treated and native fucoidan. The number of fucoidan monosaccharide units was unchanged during ultrasound treatment, meaning only chain scission was observed. This is evident in Figure 9, where the structure becomes smooth once the temperature of degraded fucoidan rises from 30 °C to 40 °C, leading to rounded and thin-walled shapes. Additionally, when the temperature increases to 50 °C, it loses its shape, forming coagulated thick-walled shapes with an uneven round circle compared to the control.

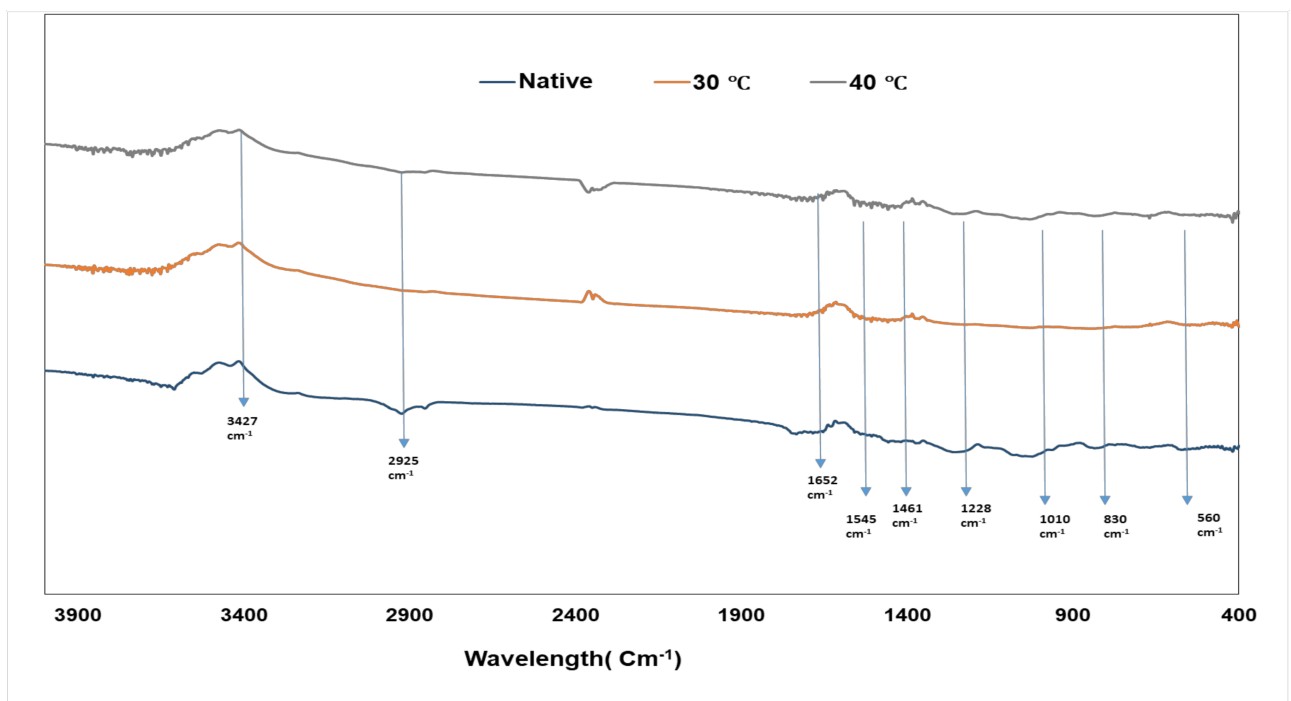

**Figure 8.** FTIR analysis of native and treated fucoidan at different temperatures.

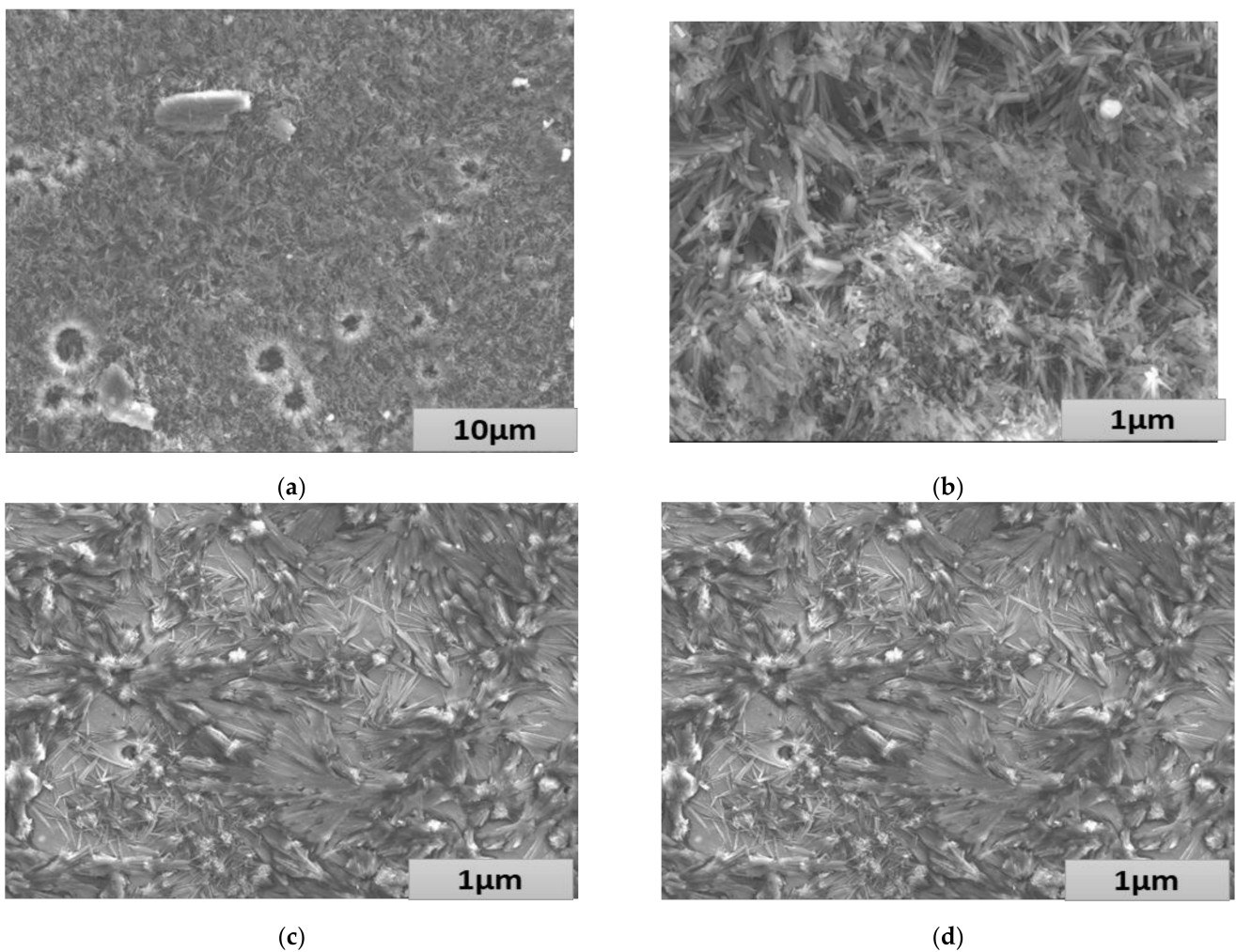

**Figure 9.** Scanning electron microscope images of the fucoidan sample (**a**) and the control at (**b**) 30 °C, (**c**) 40 °C, and (**d**) 50 °C.

### 3.7. Monosaccharide Composition and Sulfate Content

Fucose was the sole monosaccharide found in all of the fucoidan fractions according to examination of their monosaccharide composition. The native and degraded fractions of fucoidan at 15, 30, 45, and 60 min had sulfate levels of 23.12, 24.08, 24.72, 26.15, and 27.04%, respectively. A longer ultrasonic duration resulted in a modest rise in the sulfate concentration. This pattern suggested that while some slight damage to the polysaccharide chain may have occurred during the ultrasound process, the sulfate group, which is essential to bioactivity, did not sustain significant harm.

### 3.8. Mechanism of Fucoidan Ultrasonic Degradation

The ultrasonic process is a gentle, efficient, and sustainable way to break down polysaccharides. It is well accepted that ultrasonic waves may produce H• and HO• radicals, as well as shear force. According to the current theory, mechanical effects rather than radical ones are more likely to account for the ultrasonic breakdown of polysaccharides, such as in breakdown of the sulfated polysaccharide from red algae. Based on the findings of the structural study and antioxidant activity, we hypothesized that a combination of mechanical and free radical degradation may be the possible ultrasonic mechanism operating on fucoidan (Figure 10).

**Figure 10.** Mechanism of ultrasound degradation in fucoidan (1) mechanical effect broke the hydrogen bonds among the fucoidan molecules and (2) free radicals produced by ultrasonic treatment acted on the nonsulfated fucose in the backbone (A–D are repeating sulfate group).

The average Mw of fucoidan rapidly decreased during the initial stage, although no clear structural alteration was seen. In the meantime, the degraded fractions' antioxidant activity dramatically increased. As a result, we hypothesized that rather than a chemical reaction, the degradation was instead mostly caused by the mechanical impact on the dissociated aggregate polymer. The active groups associated with antioxidant activity were not harmed. Consequently, the main factor driving the average Mw to fall during this period was the broken hydrogen bonds between the fucoidan molecules. As the ultrasonication process proceeds, the polysaccharide chains are broken into a single chain and become increasingly difficult to fracture, making it difficult for the Mw to decrease. At the same time, the declining antioxidant activity is a sign that some chemical groups are being destroyed. As a result, it is challenging for the chain to be affected mechanically, and the degradation may be the result of free radicals attacking the sulfated fucose in the backbone. However, additional investigation should be undertaken to confirm the precise degradation bonds.

## 4. Statistical Analysis

Design Expert V13 was used for ANOVA for each data point, including mean standard of deviation. For statistical analysis, all samples were examined in triplicate.

## 5. Conclusions

The impact of ultrasound parameters on fucoidan from *Fucus vesiculosus* obtained from the Pacific Ocean was effectively used by the Box–Behnken design (BBD) RSM to optimize ideal conditions for lowering molecular weight with higher antioxidant activity. The ideal conditions for fucoidan degradation were a temperature of 33 °C, a sonication time of 56 min, and a sonication intensity of 116 W/cm$^2$ ($p < 0.0001$). Under ideal circumstances, the absolute errors between projected and experimental values were small, demonstrating the appropriateness of the developed models for response prediction. Under these optically ideal circumstances, molecular weight, the PDI value, and antioxidant activity were 318 kDa, 1.11, and 87.20%, respectively. Sonication treatment allowed for a limited degree of structural change in the fucoidan samples when observed in SEM and XRD analysis, which is beneficial for lowering

molecular weight and allowing oral absorption and utilization rates to be upgraded while helping the mechanism of sonication to be understood. The limitation of this work is that an excessive duration of ultrasound treatment might break the structure of fucoidan and lead to even further desulfonation occurring. Structural analysis of single-chain fucoidan extracted from treated fucoidan would be a further challenging aspect.

**Author Contributions:** U.B. was responsible for conceptualization, methodology, validation, formal analysis, investigation, resources, data curation, writing—original draft preparation, and writing—review and editing. A.K., A.M. and V.A. were responsible for formal analysis and validation. I.P. and S.H.S. were responsible for supervision and project administration. All authors have read and agreed to the published version of the manuscript.

**Funding:** The authors acknowledge RSF grant 22-76-10049 for support with experimental procedures and the characterization of materials in this manuscript.

**Institutional Review Board Statement:** There are no competing interests, and no study on animals or humans was conducted for this manuscript.

**Informed Consent Statement:** There are no competing interests, and no study on animals or humans was conducted for this manuscript.

**Data Availability Statement:** The datasets used and/or analyzed during the current study are available from the corresponding author on reasonable request.

**Conflicts of Interest:** The authors declare no conflict of interest. The funders had no role in the design of the study; in the collection, analyses, or interpretation of data; in the writing of the manuscript; or in the decision to publish the results.

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
