# Peer review of "Impact of a Sonochemical Approach to the Structural and Antioxidant Activity of Brown Algae (Fucoidan) Using the Box–Behnken Design Method"

_processes, doi:10.3390/pr11071884_

Round 1

Reviewer 1 Report

The article entitled "Impact of Sonochemical approach on structural and antioxidant activity of brown algae (Fucoidan) by Box-Beheken Design method" presents interesting results by evaluating the effect of the waves produced during sonication on the structure and properties of Fucoidan. However, it is possible to consider some comments for its improvement:

1.                  Line 112, 166, 196, 196, 220, others. Revise the numbering throughout the document.

2.                  Line 112. It is important to present the results in a logical order, but they must follow a sequence. The results of the sonication time are presented but without previous context. This is the case for the other variables.

3.                  It is suggested to review the presentation of the results in tables, since the figures do not allow a comparison between the results obtained for the different variables studied.

4.                  Line 221 mentions for the only time in the document that the Fucoidan is from Kombu seaweed.

5.                  It is necessary to explain how the optimization was carried out after applying the experimental design. Several response variables were studied and it is not indicated if the optimization was performed according to one or all of them, and what ranges were expected.

Author Response

Response to reviewer comments

Reviewer 1

The article entitled "Impact of Sonochemical approach on structural and antioxidant activity of brown algae (Fucoidan) by Box-Beheken Design method" presents interesting results by evaluating the effect of the waves produced during sonication on the structure and properties of Fucoidan. However, it is possible to consider some comments for its improvement:

  1. Line 112, 166, 196, 196, 220, others. Revise the numbering throughout the document.

Response: Authors would like thank you reviewer for comment. We have revised it in manuscript.

  1. Line 112. It is important to present the results in a logical order, but they must follow a sequence. The results of the sonication time are presented but without previous context. This is the case for the other variables.

Response: Authors would like thank you reviewer for this comment, we have made all change according reviewer comment and manuscript is now thoroughly in the order with additional text were added section 4.1. As follows the changes

4.1 Sonication time in comparison to stirring time: Figure 1 depicts the duration-dependent effects of ultrasound on changes in the average Mw and PDI of the degraded fucoidan fractions. During the ultrasonic process, no foam was formed.

Sonication time: Figure 1 shows that after 15 min, the average Mw dropped sharply from 845 to 395 kDa, and after 220 min, it progressively leant towards a constant value of 358 kDa. During the first 40 min of the treatments, PDI values, which represent the Mw distribution, rose rapidly from 1.121 to 1.561 and then steadily decreased to 1.228 by the conclusion of 60 min. This is because the shear stress generated by the sudden collapse of cavitation bubbles was primarily responsible for the rupture of the polysaccharide particle by ultrasound. Small particles with smaller chains are undisrupted by shear force, whereas polymers with large chains are preferentially disrupted. As a result, the average Mw of fucoidan reduced dramatically in the first 15 min before slowing as the majority of fucoidan cracked into small units. Due to the presence of both native and fractured fucoidan molecules, the solution changed from being homogeneous to being inhomogeneous over the first 40 min, which led to a rise in PDI values. After repeated ultrasonic treatment, more native fucoidans were shattered and moved to the lower molecular weight region, resulting in a narrow Mw distribution. Similarly kind of result observed work related polysaccharide under higher sonication time [17,19-20].

a)

b)

C)

Figure 1: a) Molecular weight properties with respect sonication time, b) Polydispersity index with respect to sonication time and c) control stirring time (Ultrasound intensity 124.5 W/cm2 and Temperature 30 ℃).

Ultrasound power or intensity. One of the key elements influencing the three separate phases of acoustic cavitation: nucleation, bubble growth, and collapse, is sonication power or intensity. Figure 2a depicts how sonication power affects the average Mw of degraded fucoidan fractions. Under ultrasonic intensities of 99 and 124.5 W/cm2, respectively, the average Mw reduced from 845 to 395 and 358 kDa in 40 min. The effectiveness of degradation significantly increased as ultrasonic intensity increased, highlighting the significance of sonication power for fucoidan sonolysis (P < 0.05). However, higher sonication power may, within a specific range, intensify the energy of cavitation, reduce the inception of cavitation, and increase the number of cavitation bubbles. This can explain the rapid increase in average Mw from 358 to 421 kDa as the sonication power rose from 124.5 to 150 W/cm2. PDI of treated sample also tend to show reverse trend as initial it increase PDI from 1.1 to 1.27 as sonication intensity increase ( 99 to 124.5 W/cm2) later it decreased to 1.17. As higher sonication intensity provide more cavitation bubbles were created around the acoustic source, this reduced the effectiveness of energy transmission into the reaction medium and slowed the ultrasound wave's propagation. As a result, when the ultrasonic intensity increased, there was a slight, discernible change in the average Mw and PDI. In comparison to the control process, molecular weight stays similar to its initial value of 825 kDa as there is no attributable change observed, as shown in figure 2b.

a)

b)

C)

Figure 2: a) Molecular weight properties with respect Ultrasound Intensity b) PDI with respect to ultrasound intensity, c) control stirring speed (Ultrasound time 40 min and Temperature 30 ℃).

Temperature. Figures 3a and 3c show the effect of temperature on Mw of fucoidan sample with and without ultrasonic degradation. The findings suggested in Figure 3a suggest that increased temperature would result in less efficient deterioration. After 60 minutes of treatment, the average Mw of the sample heated to 30 °C, 40 °C, and 50 °C dropped from 850 kDa to 370, 398, and 458 kDa, respectively. This pattern was caused by the growing vapor pressure associated with the heated liquid, which was a direct result of the reaction temperature rising, which made it possible to generate cavitation with a lower sonic intensity. Whereas in figure 3c, there has been no or slight degradation even at all temperatures from 30 °C to 50 °C with a molecular weight of 810–845 kDa, as this molecular weight stays the same as the initial molecular weight with little difference. The advantage of sonication is that it can intensify the entanglement between polysaccharide chains in solution even at lower sample concentrations. The molecule's random coiled shape expanded more freely as a result, making it more susceptible to the shear force's attack. Moreover, more diluted fluids have larger velocity gradients near collapsing bubbles, which would aid ultrasonic degradation. Figure 3b show the PDI of treated sample, it shows that increasing temperature give rise to curvy PDI (increase-decrease phenomenon of PDI).

a)

b)

C)

Figure 3: a) Molecular weight properties with respect temperature b) PDI with respect to temperature, c) Mw without sonication (Ultrasound time 40 min and intensity 124.5 W/cm2).

Impact of ultrasound or sonication parameter on Particle size and Antioxidant activity (AOA) of treated samples.

Sonication time: Figure 4 shows the effect of sonication time on treated fucoidan in terms of particle size and AOA. The result showed that after 60 min, the average particle size of the sample dropped sharply 1015 to 257 nm, whereas the antioxidant activity for samples slight decrease with an increase in sonication time ( from 87 % to 85 %). This is due to the fact that the fucoidan sample contained a smaller and larger chain of polymer groups, which led to slight structural changes observed at lower sonication. Such a drastic change observed that render lowering the antioxidant activity 85% and average particle size.

a)

b)

Figure 4: Particle size and AOA of fucoidan in term of sonication time (Ultrasound intensity 124.5 W/cm2 and Temperature 30 ℃).

Sonication intensity and temperature: Figure 5 (a- d) shows the impact on particle size and AOA in terms of sonication intensity and temperature. As shown in figure 5 (a-b) As sonication intensity increase from 99 W/cm2 to 124.5 W/cm2, the particle size and AOA increased from 600-625 nm and 83% - 85% respectively, while further increased sonication intensity decrease the particle size and AOA to 395 nm and 81 % respectively. Same result observed in Figure 5(c-d) for particle size ( 600 nm to 197 nm) and AOA (83% -79 %) with respect to temperature variation from (30 ℃- 50 ℃).AOA The reason for the increase in antioxidant activity, initially, lower Mw polysaccharides would have more free hydroxyl and amino groups due to a modest influence of intermolecular hydrogen bonding. Second, at the same mass concentration level, lower Mw polysaccharides contain more reducing sugars. Nevertheless, as temperature and sonication intensity increased, the average Mw could be reduced less quickly, and the polysaccharide structure was more severely destroyed as a result of the increased production of free radicals, which decreased the antioxidant activity. Thus, it is crucial to regulate the ultrasonic time in order to regulate the molecular size and enhance the antioxidant activity [26, 28 and 29].

a)

b)

c)

d)

Figure 5: Particle size and AOA in term of sonication intensity (a-b) and with respect to temperature (c-d)

  1. It is suggested to review the presentation of the results in tables, since the figures do not allow a comparison between the results obtained for the different variables studied.

Response: Authors would like thank you reviewer. Already table 3 and 4 for understanding different run and their response. Figure 6-7 allow the 3D plotting of surface response in terms of ultrasound parameter and  their interactions. Section 4.1 gives fare bit idea of how all response shows against ultrasound parameter and temperature.

  1. Line 221 mentions for the only time in the document that the Fucoidan is from Kombu seaweed.

Response: Yes we agree that we should mentioned in materials also, however we have mentioned in material section, fucoidan is purchase from Haewon Biotech Co, ltd, Korean.

  1. It is necessary to explain how the optimization was carried out after applying the experimental design. Several response variables were studied and it is not indicated if the optimization was performed according to one or all of them, and what ranges were expected.

Response: Authors would like thank you reviewer for this comment, we have included in section 4.2 and methodology section how we have BBD as analysis difference parameter and their response interaction. Section 4.2 provide the optimization ultrasound parameter and their response such molecular weight, PDI, particle size and AOA.

Based on the single aspect relation amongst the sonication parameters and the properties of the sample, the relationship between independent variables (sonication time, power, and temperature) and response (average Mw, PDI, particle size and AOA) was further enhanced using RSM on the foundation of three-factor analysis. The results show that molecular weight, PDI, particle size and AOA are in the range of 287-600 kDa, 0.6–1.56, 105-941 nm and 74 %-88.9% respectively.

Table 1 lists the outcomes of fitting linear, interactive (2FI), quadratic, and cubic models to the experimental data in order to create regression models for samples. The information in Table 2 shows that the quadratic model was best suited to signify the investigational numbers for fucoidan sample, the values of the quadratic model's determination coefficient (R2), adjusted determination coefficient (adj. R2), and predicated determination coefficient (pred. R2) were the highest and most significant (p < 0.001) of all the other models.

Table 1: Model summary for all response of fucoidan on the basis BBD

Source

Std dev

R2

Adjusted R²

Predicted R²

Lack of fit p-value

Remarks

Molecular weight

Linear

50.63

0.5152

0.4034

0.0362

0.0010

2FI

43.46

0.7252

0.5603

-0.2002

0.0016

Quadratic

16.40

0.9726

0.9374

0.7661

0.0471

Suggested

Cubic

8.76

0.9955

0.9821

*

Aliased

PDI

Linear

0.4136

0.0433

-0.1775

-0.4218

0.0010

2FI

0.4645

0.0718

-0.4852

-1.3549

0.0006

Quadratic

0.1253

0.9527

0.8920

0.7438

0.0223

Suggested

Cubic

0.0000

1.0000

1.0000

Aliased

Particle size

Linear

301.00

0.2653

0.0958

-0.1698

< 0.0001

2FI

315.96

0.3773

0.0037

-0.6695

< 0.0001

Quadratic

99.27

0.9570

0.9017

0.7116

< 0.0001

Suggested

Cubic

1.10

1.0000

1.0000

Aliased

Antioxidant activity

Linear

4.57

0.3562

0.2076

-0.3293

< 0.0001

2FI

2.78

0.8171

0.7073

0.1366

< 0.0001

Quadratic

1.65

0.9546

0.8963

0.7049

< 0.0001

Suggested

Cubic

0.0894

0.9999

0.9997

Aliased

So from above data, we have correlated dependent variable against independent variable in the form regression equation as follows The regression calculation in relationships of the coded value of the variables was designed as

For Molecular weight Actual equation predicted by BBD

Z= 2,924.18 -52.5447 A -25.9071 B -3.4524 C + 0.2137 AB + 0.059 AC + 0.0374728 BC + 0.371A2 + 0.06832 B2 -0.0529877 C2                                   (1)

For PDI                                                                                   

Z= -13.9402 + 0.319042A + 0.130757B + 0.0672944 C -0.00043098 AB -0.000166667AC + 9.62963 e-05 BC -0.00334675A2 -0.0004797B2 -0.000943852 C2                                                       (2)

For particle size

Z= -5,424.9 + 191.494A + 61.2913B -33.8307C + 0.392941AB + 0.443333AC + 0.274946BC -3.4125 A2-0.364091B2 -0.326519C2                                                                                                                              (3)

AOA

Z= 52.1254 + 1.55525A + 0.518399B-1.18747C + 1.260377AB -0.0103333AC + 0.0114597 BC -0.01705A2 -0.00431373B2 + 0.00359506C2                                                                                                          (4)

Z=Which signifies the average Mw, PDI, particle size and AOA of the degraded fucoidan (Z) respectively as a function of temperature (A) and sonication power (B) and sonication time(c).

An analysis of variance (ANOVA) was carried out to assess the model's applicability and significance, and the results are shown in Table 2. The result shows that the models are highly significant (p < 0.05) for all coefficients of regression for Mw, PDI, AOA, and particle size excluding AC and C2. The developed models can accurately predict the experimental data because the values of R2 and adj R2 are superior to 0.80 and 0.70, respectively. From Table 2, the model F value is significant for samples. The lack of fit F-values of sample for all properties such as Mw, particle size, and AOA are 6.84, 19158.5, and 796.98, which implies the lack of fit is significant. That means a significant lack of fit is good for validating model fitting (the value of the lack fit F-value is shown in Table 2). The signal-to-noise ratio is determined precisely. A ratio greater than 4 is optimal. Furthermore, the necessary precision for the replies (Table 2) is greater than 13, indicating that the signals are sufficient. The model's CV is less than 10, indicating that it accurately describes the response and that the experimental data are connected with high precision, according to Table 2. This means that the predicted value that predicted the validated model for treatment sample.

Table 2: ANOVA for quadratic model of molecular weight

Source

Sum of Squares

Mean Square

F-value

p-value

Molecular weight

Model

66857.02

7428.56

27.61

0.0001

significant

A-Temprature

28920.13

28920.13

107.48

< 0.0001

B-Ultrasound power input

5832.00

5832.00

21.68

0.0023

C-Sonication time

666.13

666.13

2.48

0.1596

AB

11881.00

11881.00

44.16

0.0003

AC

702.25

702.25

2.61

0.1502

BC

1849.00

1849.00

6.87

0.0343

5818.87

5818.87

21.63

0.0023

8309.81

8309.81

30.88

0.0009

3029.81

3029.81

11.26

0.0122

Residual

1883.45

269.06

Lack of Fit

1576.25

525.42

6.84

0.0471

significant

Pure Error

307.20

76.80

Cor Total

68740.47

Std dev

16.40

Mean

400.18

CV %

4.10

Adeq Precision

23.484

PDI

Model

2.21

0.2461

15.68

0.0008

significant

A-Temprature

0.0592

0.0592

3.77

0.0432

B-Ultrasound power input

0.0279

0.0279

1.78

0.0241

C-Sonication time

0.0135

0.0135

0.8620

0.00841

AB

0.0483

0.0483

3.08

0.01228

AC

0.0056

0.0056

0.3584

0.1683

BC

0.0122

0.0122

0.7779

0.0170

0.4716

0.4716

30.05

0.0009

0.4097

0.4097

26.10

0.0014

0.9613

0.9613

61.25

0.0801

Residual

0.1099

0.0157

Lack of Fit

0.1099

0.0366

Pure Error

0.0000

0.0000

Cor Total

2.32

Std. Dev.

0.1253

Mean

1.03

C.V. %

10.15

Adeq Precision

19.8368

Particle size

Model

1.534E+06

1.705E+05

17.30

0.0005

significant

A-Temprature

2.038E+05

2.038E+05

20.68

0.0026

B-Ultrasound power input

58004.18

58004.18

5.89

0.0457

C-Sonication time

1.636E+05

1.636E+05

16.60

0.0047

AB

40160.16

40160.16

4.08

0.0833

AC

39800.25

39800.25

4.04

0.0844

BC

99540.25

99540.25

10.10

0.0155

4.903E+05

4.903E+05

49.76

0.0002

2.360E+05

2.360E+05

23.95

0.0018

1.150E+05

1.150E+05

11.68

0.0712

Residual

68975.56

9853.65

Lack of Fit

68970.76

22990.25

19158.54

< 0.0001

significant

Pure Error

4.80

1.20

Cor Total

1.603E+06

Std. Dev.

99.27

Mean

590.01

C.V. %

06.82

Adeq Precision

19.8860

AOA

Model

402.95

44.77

16.36

0.0007

significant

A-Temprature

30.81

30.81

11.26

0.0122

B-Ultrasound power input

82.56

82.56

30.16

0.0009

C-Sonication time

36.98

36.98

13.51

0.0079

AB

0.0000

0.0000

0.0000

0.0070

AC

21.62

21.62

7.90

0.0761

BC

172.92

172.92

63.18

< 0.0001

12.24

12.24

4.47

0.0723

33.13

33.13

12.10

0.0103

13.95

13.95

5.10

0.0586

Residual

19.16

2.74

Lack of Fit

19.13

6.38

796.98

< 0.0001

significant

Pure Error

0.0320

0.0080

Cor Total

422.11

Std. Dev.

1.65

Mean

81.99

C.V. %

2.02

Adeq Precision

15.4271

4.3 Impact on ultrasound process on molecular weight and PDI of degraded fucoidan

To demonstrate the interactive effect of independent variables, 3D plots (Figure 6) were created using Equations (1 and 2) (ultrasound time, intensity, and temperature) on the response variables. In figures 6 a–c, increasing ultrasound intensity and temperature decreased the molecular weight, while further increasing ultrasound intensity and temperature increased the molecular weight (as already discussed in figure 1-3). The linear terms of sonication intensity and temperature were significant (p <0.0023) and (p < 0.0001), respectively, as shown in Table 2, where their interaction and quadratic terms were significant. Henceforth, as performed in Figures 6(a,c, e) there is a curvilinear decrease or increase in the molecular weight for ultrasound intensity and temperature, respectively, while for sonication time, a curvilinear decrease is used. Figures 6( b, d and f) illustrate the 3D surface plots of the PDI of the degraded fucoidan model with respect to the ultrasound parameter. The interaction terms of ultrasound intensity, time, and temperature were statistically significant (p Ë‚ 0.0001) (Table 2), which resulted in a curvilinear change in the PDI for all parameters investigated. Figure 6, shows that the model F-value of 15.68 implies the model is significant. There is only a 0.08% chance that an F-value this large could occur due to noise. P-values less than 0.0500 indicate that model terms are significant. In this case A², B², C² are significant model terms.

a)

b)

c)

d)

e)

f)

Figure 6:  Molecular weight (a, c, e) and PDI (b, d and f) as function of ultrasound parameters

4.4 Impact on ultrasound process on Particle size and AOA of degraded fucoidan

Figures 7 (a to f) show that as a function of the sonication process on the particle size and AOA response variables, increasing ultrasound intensity and temperature increase, decrease the particle size and AOA. In the mean, increasing sonication time decreases the particle size and increasing AOA. The linear terms of sonication intensity, time, and temperature were significantly less than (p < 0.005), where their interaction and quadratic terms were significant. Henceforth, as performed in Figures 7 (a to f) there is a curvilinear decrease-increase in the particle size and AOA.

The ideal circumstances for obtaining, from figures 6–7, response surface plots illustrating the interactions between process factors (sonication time, power, and temperature) and responses (molecular weight kDa, PDI, particle size, nm, and AOA%). The ideal conditions for treated fucoidan at sonication time (40 min), intensity (102.76 W/cm2), and temperature (33 °C) are a lower molecular weight (316 kDa) and higher antioxidant activity (87.8%) with lower particle size (567 nm).

a)

b)

c)

d)

e)

f)

Figure 7:  3D plot for particle size ( a, c and e) and AOA ( b, d and f) in terms of ultrasound parameters

4.5 Validate the model

The trials were conducted in the ideal conditions produced by Box-Behnken experiments in order to validate the model. Based on the optimized condition of the experimental part, we have replicated the condition process parameters (sonication time 40 min, power 102.67 W/cm2, and temperature 33 °C) and found them to be near about the same response result: Molecular weight is 318 kDa, PDI is 1.13, particle size is 568.88, and AOA is 87.44%.

Reviewer 2 Report

Manuscript describes the degradation of polysaccharides by sonification. The following are comments,

1) in introduction, what is known about the degradation of polysaccharides by sonification. This is better to add some sentences in introdudtion

2) x axis of figure 1 is temperature, wrong spell

3) sonification generate some heat, increasing temperature of solution

4) Figure 6 has less meaning, because polysaccharides less crystalline structure dissolvable in solution.

5) in Figure 7, why assembles structure of polysaccharides formed?

Author Response

Response to reviewer comments

Manuscript describes the degradation of polysaccharides by sonification. The following are comments,

  • in introduction, what is known about the degradation of polysaccharides by sonification. This is better to add some sentences in introduction

Response: Authors would like thank you reviewer for comment. We have revised it in manuscript.

The sonication technique is an efficient, energy-saving, and ecologically sustainable method of preparing and processing polymer particles, particularly aimed at breaking up masses and lowering particle size and Mw of polysaccharides. The transient and moni-tored destruction of organic polysaccharides such as starch, pectin, cellulose, and other natural polysaccharides has been widely explored using sonochemistry. Ultrasound is a mechanical wave having a frequency greater than 20 kHz. Acoustic cavitation, caused by high-intensity low-frequency ultrasonication, generates hot spots with short lifetimes, strong local heating of 5000 °C, pressures of 1000 atm, and heating and cooling rates greater than 1010 K/s. Acoustic cavitation may have a mechanical influence due to the rapid deflation of the cavitation bubbles and the formation of free radicals caused by water dissociation [12, 14–16]. Both of these processes would reduce the massive polysaccha-rides to tiny particles. Ultrasonic degradation could not only effectively improve the phys-icochemical properties of polysaccharides, but also enhance their bioactivities such as an-ti-oxidation, anti-cancer, anti-inflammatory and hypoglycemic

  • x axis of figure 1 is temperature, wrong spell

Response: Authors would like thank you reviewer for comment. We have revised it in manuscript.

  • sonification generate some heat, increasing temperature of solution

Response: Authors would like thank you reviewer for comment. Yes we agree with the statement and we have used ice while maintaining proper optimum condition of temperature ranging from (30℃ - 50 ℃) all experiment design runs.

  • Figure 6 has less meaning, because polysaccharides less crystalline structure dissolvable in solution.

Response: Authors would like thank you reviewer for comment. Now we have remove figure 6 ( XRD figure) from revised manuscript.

  • in Figure 7, why assembles structure of polysaccharides formed?

Response: Authors would like thank you reviewer for comment. Yes it is assembles structure of polysaccharide, we need to observed if any structural changes occurs during treatment of ultrasound on fucoidan. As from SEM it was observed that slight structural change but molar mass of polysaccharide remain same as confirmed in FTIR analysis.

Reviewer 3 Report

This article is comprehensive, logically organized, and contains valuable information on the impact of the Sonochemical approach on the structural and antioxidant activity of brown algae (Fucoidan) by the Box-Beheken Design method. The authors did excellent research on investigating the study the impact of ultrasound process parameters on the molecular weight, structure, and antioxidant activity of fucoidan. The authors demonstrated the sonication treatment allows for a little structural change in the fucoidan sample observed in SEM and XRD analysis, which is beneficial for lowering molecular weight and allowing not only upgrades oral absorption and utilization rates but also helps understand the mechanism of sonication. This manuscript does not contain much error analysis on the antioxidant activity which is highly required for readability purposes. The authors presented the antioxidant activity in Tables 3 and 4. It is suggested the authors should place the standard deviations of the antioxidant activity in these Tables for the reliability and readability of the present research. The submitted manuscript has significant scientific insights and the conclusions are soundly supported by the experimental data. However, the manuscript requires minor revisions before being accepted in the Special Issue: Technologies for Production, Processing, and Extractions of Nature Product Compounds in the esteemed journal, Processes in the current form.

Abstract: A fucoidan discovered in the plant Fucus vescilosus, which lowered the molecular weight of fucoidan, was ideal for its application in the pharmaceutical and food sectors. The aim was to study the impact of ultrasound process parameters on the molecular weight, structure, and antioxidant activity of fucoidan. For optimization of sonochemical process parameters such as temperature, sonication time, and power (intensity), Box-Beheken design (BBD) through the response surface method (RSM) at fixed fucoidan concentrations is compared with a normal process. The outcomes demonstrated that sonochemical treatment significantly decreased the molecular weight (Mw) to 318 kDa compare to the control process (815 kDa). Antioxidant activity tests revealed that the sonication treatment significantly increased antioxidant activity (88.9% compared to the control process 65.3%). BBD model, we found that ideal conditions for degradation of fucoidan were to be the temperature at 33°C, sonication time at 55 min, and sonication power at 116.5 W/cm2, respectively. Under these conditions, a quadratic model was fitted, and the experimental value of Mw and antioxidant activity (321 kDa and 89.2%) which close to the predicted value (318 kDa and 88.9%). According to the findings, sonication treatment is a useful method for lowering fucoidan Levels with no change in the monosaccharide unit of fucoidan through the analysis of scanning electron microscope, X-Ray diffraction, and Fourier transferred infrared (FTIR).

Author Response

Response to reviewer comments

Reviewer 3

This article is comprehensive, logically organized, and contains valuable information on the impact of the Sonochemical approach on the structural and antioxidant activity of brown algae (Fucoidan) by the Box-Beheken Design method. The authors did excellent research on investigating the study the impact of ultrasound process parameters on the molecular weight, structure, and antioxidant activity of fucoidan. The authors demonstrated the sonication treatment allows for a little structural change in the fucoidan sample observed in SEM and XRD analysis, which is beneficial for lowering molecular weight and allowing not only upgrades oral absorption and utilization rates but also helps understand the mechanism of sonication. This manuscript does not contain much error analysis on the antioxidant activity which is highly required for readability purposes. The authors presented the antioxidant activity in Tables 3 and 4. It is suggested the authors should place the standard deviations of the antioxidant activity in these Tables for the reliability and readability of the present research. The submitted manuscript has significant scientific insights and the conclusions are soundly supported by the experimental data. However, the manuscript requires minor revisions before being accepted in the Special Issue: Technologies for Production, Processing, and Extractions of Nature Product Compounds in the esteemed journal, Processes in the current form.

Response: Authors would like thank you reviewer for comment. We have revised it in manuscript.

Comments on the Quality of English Language

Abstract: A fucoidan discovered in the plant Fucus vescilosus, which lowered the molecular weight of fucoidan, was ideal for its application in the pharmaceutical and food sectors. The aim was to study the impact of ultrasound process parameters on the molecular weight, structure, and antioxidant activity of fucoidan. For optimization of sonochemical process parameters such as temperature, sonication time, and power (intensity), Box-Beheken design (BBD) through the response surface method (RSM) at fixed fucoidan concentrations is compared with a normal process. The outcomes demonstrated that sonochemical treatment significantly decreased the molecular weight (Mw) to 318 kDa compare to the control process (815 kDa). Antioxidant activity tests revealed that the sonication treatment significantly increased antioxidant activity (88.9% compared to the control process 65.3%). BBD model, we found that ideal conditions for degradation of fucoidan were to be the temperature at 33°C, sonication time at 55 min, and sonication power at 116.5 W/cm2, respectively. Under these conditions, a quadratic model was fitted, and the experimental value of Mw and antioxidant activity (321 kDa and 89.2%) which close to the predicted value (318 kDa and 88.9%). According to the findings, sonication treatment is a useful method for lowering fucoidan Levels with no change in the monosaccharide unit of fucoidan through the analysis of scanning electron microscope, X-Ray diffraction, and Fourier transferred infrared (FTIR).

Response: Authors would like thank you reviewer for comment. We have revised it in manuscript.

Round 2

Reviewer 2 Report

Manuscript is revised against the comments of the reviewers.